# Geothermal heat flow in Antarctica: current and future directions

Alex Burton-Johnson[1], Ricarda Dziadek[2], and Carlos Martin[1]

[1]British Antarctic Survey, High Cross, Madingley Road, Cambridge, CB3 0ET, UK

[2]Alfred Wegener Institute - Helmholtz Centre for Polar and Marine Research, Am Alten Hafen, Bremerhaven, Germany

*Correspondence to:* Alex Burton-Johnson (alerto@bas.ac.uk)

## 1. Abstract

Antarctic geothermal heat flow (GHF) affects the temperature of the ice sheet, determining its ability to slide and internally deform, as well as the behaviour of the continental crust. However, GHF remains poorly constrained, with few and sparse local, borehole-derived estimates, and large discrepancies in the magnitude and distribution of existing continent-scale estimates from geophysical models. We review the methods to estimate GHF, discussing the strengths and limitations of each approach, compile borehole and probe-derived estimates from measured temperature profiles, and recommend the following future directions: 1) Obtain more borehole-derived estimates from the subglacial bedrock and englacial temperature profiles. 2) Estimate GHF from inverse glaciological modelling, constrained by evidence for basal melting and englacial temperatures (e.g. using microwave emissivity). 3) Revise geophysically-derived GHF estimates using a combination of Curie depth, seismic, and thermal isostasy models. 4) Integrate in these geophysical approaches a more accurate model of the structure and distribution of heat production elements within the crust, and considering heterogeneities in the underlying mantle. And 5) continue international interdisciplinary communication and data access.

## 1.  Introduction

The Antarctic ice sheet is the world's largest potential driver of sea level rise, and accurately modelling its dynamics relies, amongst others, on constraining conditions at the ice-bedrock interface. Measuring these basal conditions is inherently challenging and, of all the parameters affecting ice sheet dynamics, subglacial geothermal heat flow (GHF) is one of the least constrained (Larour et al., 2012; Llubes et al., 2006). Despite this uncertainty, GHF affects (1) ice temperature and, as a consequence, ice mechanical properties (rheology), (2) basal melting and sliding, and (3) the development of unconsolidated water-saturated sediments; all of which can promote ice flow (Greve and Hutter, 1995; Larour et al., 2012; Siegert, 2000; Winsborrow et al., 2010). Beyond ice dynamics, our knowledge of GHF allows us to model past and present basal melt rates in our exploration for old ice core climate records (Van Liefferinge et al., 2018), constrain models of glacial isostatic adjustment (GIA; van der Wal et al., 2013, 2015), and inform on the geological and tectonic development of Antarctica (McKenzie et al., 2005).

In recognition of the ambiguity and importance of Antarctic GHF, an increasing number of studies in geology, geophysics, and glaciology have sought to constrain this parameter, with a developing dedicated multinational interdisciplinary community (Burton-Johnson et al., 2019; Halpin and Reading, 2018). However, with an expanding research base and a requirement for multidisciplinary science, the necessity for a multidisciplinary review of current approaches and future directions was highlighted by the GHF sub-group of SERCE (Solid Earth Response and influence on Cryospheric Evolution) and the Scientific Committee on Antarctic Research (SCAR) (Burton-Johnson et al., 2019). This paper also provides the background material for a SCAR-commissioned White Paper on future research directions (Burton-Johnson et al., 2020).

### 1.1.  What is geothermal heat flow (GHF)?

GHF describes the transport of heat energy from the interior of the Earth to the surface (Gutenberg, 1959; Pollack et al., 1993). This heat originates from two primary sources: 1) The primordial heat remaining from the formation of the Earth, when the kinetic energy of celestial collisions was transformed into heat energy; and 2) the radioactive decay of heat-producing elements (HPEs) and their isotopes; 98% of which is derived from Uranium, Thorium, and Potassium (Beardsmore and Cull, 2001; Lowrie, 2007). The HPEs are incompatible with the mineral structures of the mantle, so are concentrated into the crust (Boden, 2016; McDonough and Sun, 1995). Other sources of possible contributions to GHF are: 1) geoneutrino emission from the mantle (Huang et al., 2013; Korenaga, 2011), and 2) gravitational pressure (Elbeze, 2013; Morgan et al., 2016).

The estimated average heat flow of continental crust is 67.1 mW m$^{-2}$, whilst for oceanic crust it is 78.8 mW m$^{-2}$ (Lucazeau, 2019; although estimates vary according to sampling strategy and the number of observations). The difference between continental and oceanic heat flow reflects the smaller thickness of oceanic crust, with hot mantle rocks at comparatively shallow depths. Continental GHF varies significantly, primarily in response to variations in crustal heat production, age, composition, tectonic history, and thickness of crust and mantle (Mareschal and Jaupart, 2013). This results from the geological complexity of composite continental crust compared with oceanic crust. GHF is generally lower in stable crust away from convergent and divergent continental margins and rift basins, and higher in these magmatically active provinces (Lucazeau, 2019; Pollack et al., 1993). On a broad regional scale, continental GHF correlates negatively with age, allowing first order empirical estimation of Antarctic GHF based on its range of crustal ages (Fig. 1; Llubes et al., 2006; Sclater et al.,

1980). However, Antarctic crustal heat production estimates show high variability across sampled age ranges
(Gard et al., 2019), with lithology and tectonic setting being important controls on the heat production distribution
(Carson et al., 2014; Halpin et al., 2019).

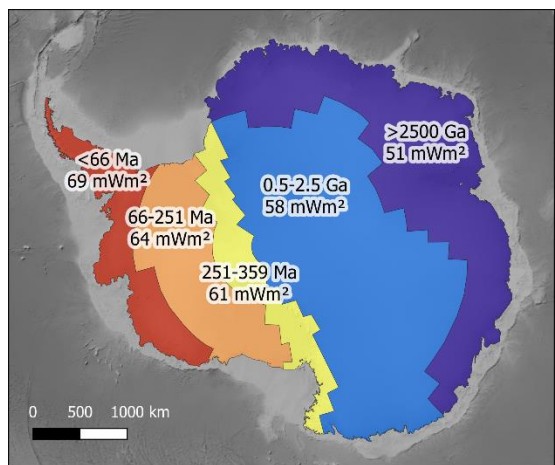

**Fig. 1. Empirical estimation of GHF based on generalised Antarctic crustal ages and mean global GHF values of**
**continental crust of similar age (adapted from Llubes et al., 2006). Basemap bathymetry from ETOPO1 (Amante and**
**Eakins, 2009).**
The rate of heat flow, $Q$, can be approximated by the Fourier's Law (Baron Fourier, 1822). In the simple model
of a homogenous material with a constant thermal gradient, this equates to:
$$Q = -\kappa \, \partial T / \partial z$$

70 (1)

Where $Q$ has the units mW m$^{-2}$ (i.e. power per unit area); $T$ is the temperature (K), $z$ is the vertical distance (m);
and $\kappa$ is the thermal conductivity of the material (mW m$^{-1}$ K$^{-1}$). When considering the basal conditions of the
Antarctic ice sheet, we are interested in the heat flow at bedrock surface. We also need to consider internal heat
production, $A$ ($\mu$W m$^{-3}$). For a simple case of constant thermal conductivity and heat production, surface heat flow
can be described by:
$$Q = \kappa_d [\partial T / \partial z]_d + \int A(z) \delta z$$

77 (2)

Where the integral is measured from the surface to a depth, $d$ (equation 1.13 in Beardsmore and Cull, 2001).
We would like to highlight here that most methods to estimate GHF derive it from the temperature gradient, as in
Equations 1 and 2. However, these equations are a simplification, as temperature variation over time, surface
topography, internal heat production, and variation in the properties of the material all affect the observed
temperature gradient.

 1.2. **A note on terminology: Heat "Flow" vs Heat "Flux"**

In the scientific literature, heat "flow" and heat "flux" are used interchangeably. The consensus from the SCAR-
SERCE White Paper authorship (Burton-Johnson et al., 2020) is that "flow" is the correct terminology. "Heat
flow" is not limited to the movement of material, but the mechanism of heat transfer (dominantly by conduction
when near the Earth's surface). Although the two terms are used interchangeably, heat "flow" has been established
for decades to describe the rate of heat transferred across the surface of Earth per unit area, is the term used by the
International Heat Flow Commission, and is thus the term used here. We recommend adopting this term in
preference in the future, although the most important consideration is to state the correct units (mW m$^{-2}$).
**2. Motivation: What is the importance of GHF in Antarctica?**
2.1. **Glaciology**
GHF can strongly influence the basal temperature of the ice sheet. As a consequence, it is a key contributor to
basal meltwater production, ice rheology, basal friction, basal sliding velocity, and erosion (Fahnestock et al.,
2001; Goelzer et al., 2017; Hughes, 2009).
The heat budget at the base of an ice sheet can be described (Vieli et al., 2018):
$$Q_g + Q_s + Q_w + Q_p + Q_f + Q_c = 0$$

98 (3)

Where $Q_g$ is the GHF, $Q_s$ is the heat generated by sliding, $Q_w$ is the heat generated by subglacial water flow, $Q_p$
is the heat required to maintain the flowing water at pressure melting point, and $Q_f$ is the heat released by freezing
or used by melting; and $Q_c$ is the heat conducted away in the ice towards the ice surface. Of the positive
contributions to basal heat, that generated by sliding ($Q_s$) can be orders of magnitude greater than that from GHF
($Q_g$), but in slow flowing areas $Q_s$ is negligible and GHF plays a key role in the heat budget (Larour et al., 2012;
Pittard et al., 2016a).
To illustrate this point, Llubes et al. (2006) modelled a 20 mW m$^{-2}$ increase in GHF across the Antarctic continent
(from uniform values of 40 to 60 mW m$^{-2}$). This resulted in a 6°C increase in the mean basal temperature, from -
13°C to -7°C, and expanded the proportion of the basal ice area above the pressure melting point (PMP) from
16% to more than 50%. This variation directly affects the basal melt rates, with a uniform 40 mW m$^{-2}$ generating
6.7 km$^3$ yr$^{-1}$ of basal melting across Antarctica, whilst 60 mW m$^{-2}$ would generate 18 km$^3$ yr$^{-1}$. However, unlike
the GHF values used, the resultant basal temperature variation is non-uniform: Although the two heat flow models
produce only a few °C difference in basal temperature near the coast, they generate up to 15°C difference in
central East Antarctica. This is because horizontal advection and frictional basal heating are negligible beneath
the thick, slow moving ice of East Antarctica, and surface temperatures have a reduced effect on basal conditions
(Llubes et al., 2006; Pollard et al., 2005). In these regions of thick ice, the increased pressure brings the basal ice
temperature closer to its PMP (Pollard et al., 2005), and the thicker ice has a greater insulating effect. Although
the effect of pressure of basal temperature is much smaller than surface temperature variation, in areas where of
thick ice where the basal temperature is close to the PMP, even small variation in GHF can determine whether
basal melting occurs. This has a resultant effect on the basal friction and sliding of the ice sheet (Pollard et al.,
2005). In addition, the increased ice temperature makes it more susceptible to internal deformation, which also
enhances its ability to flow (Llubes et al., 2006).
Even beneath the comparatively thinner ice of West Antarctica, the sensitivity of basal temperature to heat flow
is enhanced (Llubes et al., 2006). There is evidence that this region, dominated tectonically by the West Antarctic
Rift System (Jordan et al., 2020), exhibits very high values of basal heat flow and resultant basal melting
(Schroeder et al., 2014). Above 85 mW m$^{-2}$, the basal temperature of much of the West Antarctic Ice Sheet will
pass its pressure melting point (in agreement with radar evidence for extensive basal melting; Llubes et al., 2006;
Rémy and Legresy, 2004; Schroeder et al., 2014). Consequently, enhanced basal heat flow in West Antarctica can
have a large effect on its basal melt rates, although the thinner ice sheet in West Antarctica compared to East
Antarctica makes it more sensitive to surface parameters (advection and conduction of the surface temperature,
itself influenced by the accumulation rate; Llubes et al., 2006).
In addition to enhancing basal melting and reducing basal friction, increased GHF enhances ice flow by increasing
the englacial temperature and thus reducing the ice stiffness (Larour et al., 2012). Because the heat produced by
basal friction and viscous deformation can be orders of magnitude greater than from GHF in fast-flowing ice
streams, this effect is only significant in upstream, slow-flowing areas (Larour et al., 2012). In these regions of
thick, slow-flowing ice, even local high heat flow anomalies of insufficient heat for basal melting can result in the
development of accelerated, channelised flow for hundreds of kilometres upstream and downstream of the GHF
anomaly through the effect of GHF on the ice rheology (Pittard et al., 2016a). Regions along ice divides and
adjacent to ice streams are particularly sensitive to enhanced GHF (Pittard et al., 2016b).
Whilst the points above highlight the necessity of estimating Antarctic GHF, it is very important that the accuracy
of these estimates can be verified. The impact of inaccurate GHF constraints on models of ice sheet dynamics
have been shown by comparing GHF estimates for Greenland. Ice sheet modelling controlled by spatially variable
GHF forcing reproduces the observed state to only a limited degree, and fails to reproduce either the topography
or the low basal temperatures measured in southern Greenland (Rogozhina et al., 2012). Instead, an unrealistic
spatially uniform GHF forcing produces a considerably better fit. If the much larger Antarctic ice sheet is to be
accurately modelled, the accuracy of the GHF estimates used must be well constrained by multiple independent
methodologies, sensitivity tests, and comparison of different models.
Recently, there has been increasing interest in the exploration of suitable locations for coring Antarctica's oldest
continuous ice record (Fischer et al., 2013). This problem requires accurate knowledge of GHF, as basal melt rates
limit the maximum possible age of recoverable ice (Van Liefferinge et al., 2018). Additionally, due to
environmental concerns around possible drilling fluid contamination, frozen bed conditions are a prerequisite for
deep coring operations for recovery of the oldest ice records.
**2.2. Glacial Isostatic Adjustment (GIA)**
The temperature of the lithosphere and upper mantle are important parameters for modelling the isostatic response
to changes in the volume of the overlying ice sheet (i.e. glacial isostatic adjustment, GIA). This is because the
(visco-)elastic properties of the lithosphere and mantle directly relate to its thermal properties (Chen et al., 2018;
Kuchar and Milne, 2015). GIA is a critical component of the long-term evolution of ice sheets and could
potentially stabilise retreating ice streams in submarine settings (Barletta et al., 2018; Kingslake et al., 2018). Of
particular importance here is that the temperature-dependant viscosity that controls GIA can be modelled using
surface heat flow estimates (van der Wal et al., 2013, 2015).
2.3. **Geology and tectonics**

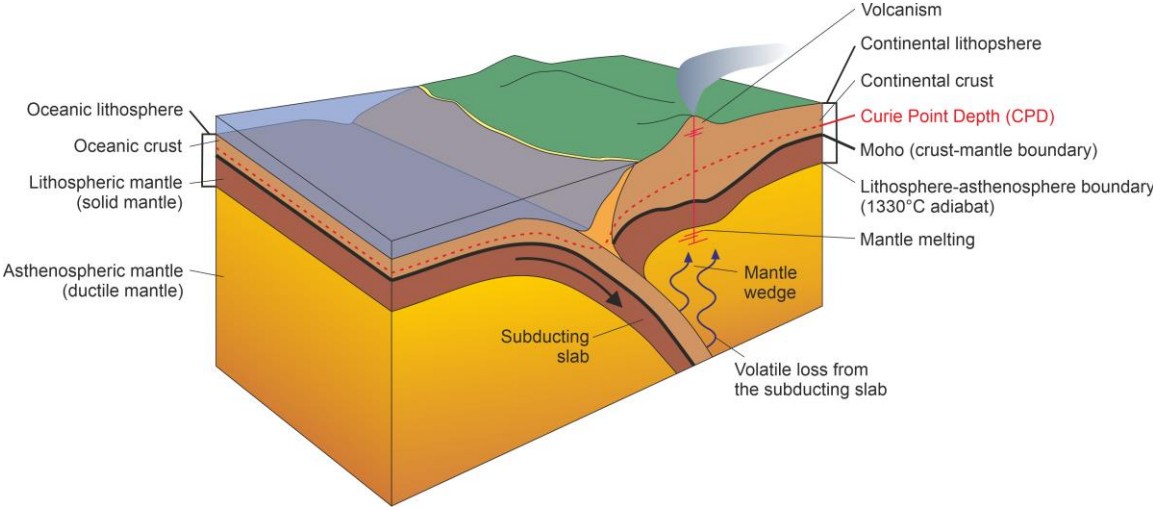


**Fig. 2. Basic illustration of a subduction zone at a convergent margin between oceanic and continental lithosphere to**
**clarify the geological concepts and terms used in this paper.**

### 2.3.1. Mantle dynamics

Heat flow variation and its isostatic effects (i.e. the buoyancy control on crustal elevation, resulting from the
different densities of the dense mantle and less dense overlying crust) provide evidence for mantle dynamics
beneath a continent. For example, high heat flow anomalies have been proposed as evidence for sub-lithospheric
heating by present and past mantle plumes (regional hot spots of warm mantle upwelling beneath the lithosphere;
e.g. Courtney and White, 1986; Martos et al., 2018), and the absence of enhanced heat flow where mantle ascent
is proposed has been used to argue against such processes (e.g. Stein and Stein, 2003). Also, because of the
relationship between surface heat flow and isostatic elevation, heat flow studies can reveal thermal or
compositional variation of the sub-continental mantle, as a reduction in its density can increase the isostatic
elevation of the surface topography (Hasterok and Gard, 2016).

### 2.3.2. Development of the lithosphere

The thermal properties of the lithosphere control its response to tectonic deformation (e.g. Sandiford and Hand,
1998), such as the development of crustal shear zones and earthquakes. The lithosphere's thermal properties also
affect the relative density of lithosphere and underlying mantle, and (as a result of this buoyancy effect) the
isostatic surface elevation. This in turn influences the heights of Antarctica's mountain ranges and the depths of
its sedimentary basins (McKenzie et al., 2005). For these reasons, understanding the continent's GHF will inform
on the development of many of Antarctica's largest tectonic features. For example, the lithospheric extension of
the West Antarctic Rift System, the prominent elevation of the Transantarctic Mountains, the deep topographic
depression of the Wilkes subglacial basin, and the extensive Palmer Land Shear Zone of the Antarctic Peninsula.

| Section | Method | Description | Advantages | Disadvantages |
|---|---|---|---|---|
| **3.** | **Measured Temperature Gradients** | | | |
| 3.1. | Bedrock boreholes | GHF estimated from measured temperature gradient into bedrock boreholes. | - Local estimates of GHF derived directly from the bedrock. | - Only point estimates.<br>- Affected by local variation.<br>- Requires drilling through the ice sheet and deep enough into the rock. |
| 3.2. | Ice boreholes | Temperature gradient measured in ice boreholes, and GHF estimated from the basal temperature gradient or models of the temperature profile. | - Provides local estimates of GHF beneath the ice sheet without drilling into bedrock. | - The ice sheet must be frozen to the bed and thermally equilibrated.<br>- Limited by modelling accuracy of the ice sheet thermal history<br>- Hot water drilling requires 2 years for thermal equilibration after drilling. |
| 3.3. | Marine/onshore sediment temperature probes | GHF estimated from shallow (<10m) temperature gradient measured using gravity-driven probes. | - Faster acquisition than borehole estimates, as no drilling required. | - Requires deep water without long-period temperature variation. |
| **4.** | **Geophysical and geological methods** | | | |
| 4.1. | Magnetic-derived estimates | Temperature gradient calculated by estimating the depth of the Curie isotherm from magnetic anomalies. | - Allows continent-scale estimates.<br>- Does not require models of the crust and mantle structure. | - Assumes that the depth to the bottom of the magnetic source is temperature controlled (i.e. it represents the Curie isotherm), despite possible other geological controls.<br>- Spatial resolution limited by altitude of sensor and depth of magnetic source. |
| 4.2. | Seismic-derived estimates | Calculate GHF empirically or via forward modelling using the relationship of mantle seismic velocity and temperature, and estimation of lithospheric thickness. | - Allows continent-scale estimates.<br>- Empirical models utilise well-constrained regions.<br>- Forward models estimate the geological source of GHF. | - Empirical estimates assume global comparison is valid.<br>- Forward models assume mantle and crustal composition.<br>- Limited spatial resolution. |
| 4.3. | Gravity model-derived estimates | Calculates GHF from models of crust and mantle structure derived from gravity estimates of crustal thickness. | - Allows large-scale GHF estimates.<br>- Incorporates constraints on crustal composition. | - Models are non-unique, requiring further constraints.<br>- Assumes values of crustal and mantle composition. |
| 4.4 | Conjugate margin-derived estimates | Reconstruct the Gondwana supercontinent, interpolating Antarctic GHF from better constrained adjacent continents. | - Utilises regions where GHF is better constrained.<br>- Should be most accurate around continental margins. | - Poor constraints away from continental margins.<br>- Affected by choice of input data and interpolation method. |
| 4.5. | Isostatic elevation | Calculates GHF from topography using a compositional correction. | - The topographical input is a well constrained variable. | - Requires assumptions of crustal thickness, density, heat production, and thermophysical properties.<br>- Low spatial resolution. |
| 4.6. | Incorporating heterogeneous crustal compositions. | Incorporating measurements of crustal heat production and models of heterogeneous crustal structure into geophysical GHF models. | - A more realistic representation of the geological sources of GHF.<br>- Reflects the concentration of heat production in the crust. | - Requires assumptions of the subglacial geology away from outcrops.<br>- The 3D structure and composition of the crust and mantle is ambiguous. |
| **5.** | **Glaciological methods** | | | |
| 5.1. | Subglacial water | Radar evidence for subglacial water used to model the required GHF distribution for required basal melting and hydrology. | - Based on observable effects of GHF. | - Requires accurate ice sheet thermal models.<br>- Subglacial water only accumulates in appropriate topographic depressions. |
| 5.2. | Subglacial lakes | Lakes identified by enhanced radar reflectivity, and the minimum GHF required for basal melting estimated from ice sheet thermal models. | - Based on observable effects of GHF.<br>- Where the ice sheet is frozen to the bed, maximum GHF can be calculated. | - Requires accurate ice sheet thermal models.<br>- Subglacial water only accumulates in appropriate topographic depressions. |
| 5.3. | Englacial stratigraphy | Melt rates and required GHF calculated from englacial layers identified in radar data. | - Based on observable effects of GHF.<br>- Identifies high GHF anomalies. | - Requires accurate data and interpolation from ice cores.<br>- Requires accurate ice sheet thermal models. |
| 5.4. | Microwave emissivity | Englacial temperatures modelled with variable GHF to simulate observed satellite-derived, temperature dependent microwave radiation. | - Derives more extensive englacial temperature profiles than can be achieved by boreholes. | - Only applicable to areas of thick, slow flowing ice.<br>- Method requires further validation. |

**Table 1. Summary of methods to estimate GHF and their section in the manuscript.**

### 3. GHF estimates from measured temperature gradients

Having highlighted the importance of constraining Antarctica's GHF, the following sections discuss current approaches to its estimation. The methods discussed are summarised in Table 1.

Local heat flow estimates can be derived by measuring the temperature at various depths below the surface (either in the bedrock, overlying sediments, or within the ice sheet) and deriving a temperature gradient. In Antarctica, GHF has been derived through temperature measurements from boreholes into the bedrock or into the ice sheet, and also from probes into unconsolidated sediments. It is important to recognise that these are "estimates" not "measurements" of GHF, particularly when using them to verify the accuracy of geophysical or inverse GHF estimates. This is because the measured thermal gradient can be affected by processes other than GHF, including surface temperature variation and hydrothermal circulation. When evaluating a specific local estimate, its derivation, local geology, and other regional GHF estimates must be considered. Thermal gradients and surface heat flow may vary significantly over 10 km lateral spatial resolutions (Carson et al., 2014) with variations in geology (affecting heat production and conductivity; Carson et al., 2014; Hasterok and Chapman, 2011), hydrothermal circulation (affecting local heat convection and redistribution; Fisher and Harris, 2010), and topography (affecting heat diffusion pathways to the surface; Bullard, 1938; Lees, 1910).

### 3.1. Boreholes into bedrock

The thermal gradient can be determined by measuring the temperature variation at different depths in the crust. Away from Antarctica, these measurements are from boreholes (commonly those drilled for mineral or hydrocarbon exploration), mineshafts, caves, or other cavities. The temperature gradient of the crust's uppermost 10-50 m is dominantly affected by downward conduction of the surface temperature rather than GHF. To address this, temperature measurements are made over the largest depth range possible (typically 100-1000 m).

Borehole temperature measurements are made using wire-line temperature probes, with a thermistor at the leading tip and measurements made progressively downwards to minimise disturbance of the borehole fluids prior to temperature measurement. The temperature is measured from the bore fluid, not the surrounding rock, so an important consideration is the need for thermal equilibration of the wall rock and the borehole fluids following drilling and prior to measurement. In addition, the heat produced during drilling needs to be dissipated from the borehole. As a guide, 10-20 times the drilling time is required before a borehole is equilibrated to within instrument accuracy (Bullard, 1947; Jaeger, 1956), although observations show that after 3 times the drilling time, borehole fluids are within 0.05°C of equilibrium values (Lachenbruch and Brewer, 1959). As an example of the time required for bedrock drilling, drilling of the multiple Cape Roberts Project boreholes averaged 16-31 m day$^{-1}$ (Talalay and Pyne, 2017). For the low water flows used in small-core (<4 cm diameter) diamond drilling (compared with the high water flows of wider core diameter rotary drilling), heat exchange is negligible except for the upper and lowermost ~20 % of the borehole, and full temperature profile measurements can be taken about two days after drilling cessation (Jaeger, 1961, 1965).

Depth below the bedrock surface must be considered when taking borehole temperature measurements. Where terrestrial bedrock is exposed, atmospheric temperature and seasonal variation perturbs the thermal gradient in the upper >100 m of the crust. In Antarctica, temperatures from Hole 3 of the Dry Valley Drilling Project provided estimates of "equilibrium" gradient only when deeper than 90 m (Decker, 1974; Decker et al., 1975; Pruss et al.,

1974). It may be possible to compensate for seasonal variation in shallower boreholes using long-term
observations of the temperature gradient (>1 year), although the previous attempt (from a 7.6 m borehole at
McMurdo Station; Risk and Hochstein, 1974) derived an anomalously high GHF estimate (164 mW m$^{-2}$, compared
to 66 mW m$^{-2}$ from a 260 m deep borehole; Decker and Bucher, 1982).
Subglacial bedrock is not exposed to atmospheric temperature variation, so the geothermal gradient can be
measured from shallower depths. However, it is affected by heat derived from the overlying ice sheet: internal
and basal frictional shear heating from the ice sheet, heat advection, basal water, and seasonal temperature
variation (e.g. Ritz, 1987). In the absence of a deep, borehole-derived, subglacial bedrock temperature profile, the
depth required to accurately measure the unperturbed geothermal temperature gradient is currently unknown.
Thermal diffusion modelling over timescales of low frequency climate variation may constrain this.
3.2. **Ice boreholes**
Subglacial GHF can be estimated from the temperature gradient from boreholes into the ice sheet (e.g. Engelhardt,
2004; Fudge et al., 2019; Nicholls and Paren, 1993). This requires that there is no additional heating from basal
shear or horizontal advection, and that the ice sheet has been unequivocally frozen to the bed for long enough that
the bedrock and overlying ice sheet have thermally equilibrated. To meet this requirement, the temperature profile
is best measured from cores into the summits of ice domes where the ice sheet is stationary (Engelhardt, 2004).
As applies to bedrock boreholes, a delay between drilling and temperature measurement is required for the thermal
disturbance from the drilling to dissipate. For hot-water drilling, this can take 2 years (Barrett et al., 2009;
Engelhardt, 2004). The temperature profile is typically measured using thermistors, recording the temperature
through changes in resistivity to electrical currents. Either a string of thermistors is deployed into the borehole
prior to freezing, and the temperature recorded over time, or the hole can be kept open with drill fluid and
downhole temperature measured with a moving thermistor. More recently, temperature has been recorded also
using distributed temperature systems (DTS; Suárez et al., 2011; Ukil et al., 2011). The temperature is derived
from the travel time of a laser beam within an optical fibre. All of these methods require thermal equilibration.
Once the englacial temperature profile is obtained, GHF estimation can be achieved through three methods.
Firstly, if the borehole reaches the ice-bedrock interface, and the bedrock and overlying ice are in thermal
equilibrium, then the GHF can be estimated in the same way as for bedrock boreholes (e.g. Engelhardt, 2004).
That is, using the temperature gradient in the ice near the ice-bedrock interface but using the thermal conductivity
of ice rather than rock (Equation 1). Secondly, rather than measuring a temperature profile above the bed, the
basal temperature at the ice-bedrock interface can be measured, and temperature modelled through time to
constrain the required GHF (e.g. Fudge et al., 2019). Thirdly, if the borehole doesn't reach bedrock, and similarly
to the previous method, a thermal model is required to constrain GHF (e.g. Zagorodnov et al., 2012). In the
methods where modelling is required, the variables are modified within constraints determined for the location
until the modelled temperature profile best fits the measurements (Fig. 3), and the modelled temperature gradient
within the bedrock used for GHF calculation.

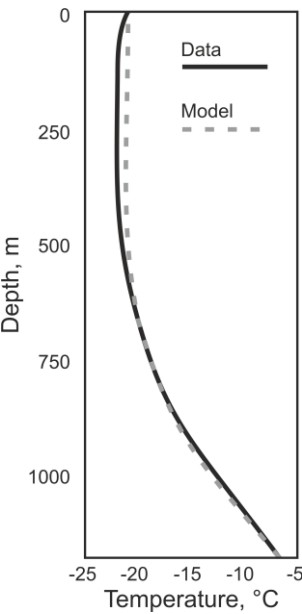


**Fig. 3. An example of temperature measurements (solid black line) and steady state model (dashed grey line) from which GHF can be estimated. Adapted from (Dahl-Jensen et al., 1999) for Law Dome ice borehole temperature profile. Note that it is the deeper temperature gradient that is modelled rather than the shallower temperature variation.**

In regions where the ice sheet is frozen to the bed and thermally equilibrated, GHF can be estimated from boreholes that do not reach the bedrock providing that the temperature profile is obtained below the penetration depth (or skin depth, $\delta$) of surface temperature variation into the ice sheet. This depth is defined by the circular frequency of the variation ($\omega$), and the thermal diffusivity of the material ($k$) according to Equation 4 (Fig. 4; Carslaw and Jaeger, 1959; Wangen, 2010).

$$\delta = \sqrt{(2\,k/\omega)}$$

(4)

Where circular frequency ($\omega$) is defined by Equation 5, where $t_p$ is the time for one period (or cycle) of the temperature variation (Wangen, 2010).

$$\omega = 2\,\pi/t_p$$

(5)

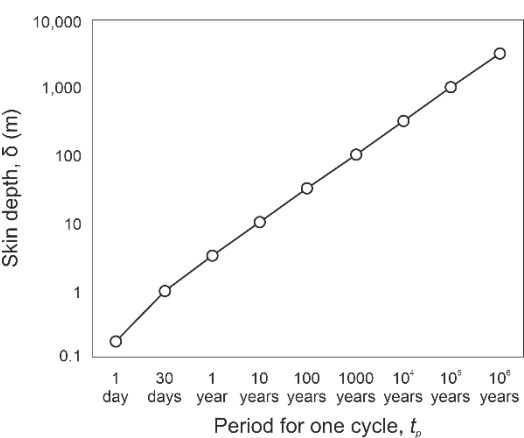

272

**Fig. 4. Relationship between skin depth and periodicity of temperature variation through a material of thermal diffusivity, $k$, of $10^{-6}$ m$^2$ s$^{-1}$. This diffusivity is comparable to ice at -10°C (James, 1968), or average values of a range of rock types at ~50°C (Vosteen and Schellschmidt, 2003), and increases with decreasing temperature for both materials.**

The deepest significant perturbations of the englacial temperature profile are from glacial-interglacial cycles, and GHF is best estimated from the englacial temperature profile below the depth at which this effect becomes negligible. In Greenland, this is the bottom 20 % of the ice sheet, but in areas of low-accumulation in Antarctica this can extend to much shallower depths. With sufficiently accurate temperature measurements, the full temperature profile of the ice sheet and the subglacial GHF may be estimated from boreholes penetrating only the upper 600 m or 20 % of the total ice sheet thickness (Hindmarsh and Ritz, 2012; Mulvaney et al., 2019; Rix et al., 2019). However, use of shallow boreholes to estimate GHF use simplified thermal models and assumptions on ice sheet evolution, and so require further validation.

However, poorly-constrained thermal effects within the ice sheet propagate uncertainties in GHF estimates from ice sheet boreholes (Cuffey and Paterson, 2010, Chapter 9). This is a particular problem if there is any ambiguity as to whether the ice sheet is frozen to the bed. The englacial temperature profile depends on heat sources at the surface, base, and within the ice (i.e. internal deformation-derived frictional heating). Heat sources that act at the base of the ice, such as frictional heating by basal motion, are impossible to differentiate from GHF.

3.3. **Marine and onshore unconsolidated sediments**

Shallow (<~10 m) temperature gradients in unconsolidated sediments can be recorded using gravity-driven probes rather than drilled boreholes. They carry multiple thermistors along the length of the probe that provide a temperature profile. These measurements can be taken from unconsolidated sediments offshore (e.g. Dziadek et al., 2019, 2017), in subglacial lakes (Fisher et al., 2015) or below ice shelves (Begeman et al., 2017).

As applies to borehole measurements, temperature gradients in unconsolidated sediments must be taken at sufficient depth to represent the crustal temperature gradient and not be perturbed by temperature variation in the overlying water or ice (i.e. they must be representative of steady-state conditions). The penetration depth of temperature variation is dependent on its frequency (Equation 4 and Fig. 4; Carslaw and Jaeger, 1959). Consequently, diurnal or annual cycles only affect the upper few centimetres to couple of metres of the surface temperature profile, whilst variations over the last 200-300 years will affect the upper 200 m, and post-glacial warming can be observed down to 2500 m. These effects are dampened by an overlying water column or ice sheet, but temperature variation over 10 kyr can still affect basal ice sheet temperatures (Engelhardt, 2004). Although large (>10 °C) seasonal temperature variations are dampened by ~90% at water depths of 3-5 m (Müller et al., 2016), long-term variations (e.g. climate-controlled variations in Circumpolar Deep Water over the last ~12 kyr; Hillenbrand et al., 2017) are likely recorded in the upper 3 m at 400 m water depth, 2 m at 700 m depth, and even the upper ~1 m at 1000 m depth (Dziadek et al., 2019).

Similarly to borehole temperature measurements, a time delay must be considered between penetration of the sediments and temperature measurement. A ten minute delay between sediment penetration and measurement is

sufficient to allow decay of frictional heating, as the temperature decay takes ~100 s (Dziadek et al., 2019; Pfender
and Villinger, 2002).
Unconsolidated temperature measurements can also be taken from marine boreholes (e.g. IODP boreholes). For
bedrock boreholes, a delay is required between drilling and measurement for thermal equilibration of the wall
rock and the borehole fluids, which would be problematic for marine boreholes where a drill ship cannot remain
on site. Instead, for boreholes into unconsolidated sediments, a probe is deployed into the borehole bottom
sediments shortly after drilling. Although technology has improved (Davis et al., 1997; Heesemann et al., 2006),
measurements can be affected by frictional heating during and after probe deployment, or by movement of water
and sediments within the hole. Only measurements that exhibit the expected temperature decay rate after
penetration are thus reliable (Hyndman et al., 1987).

## 4.    Geophysical and geological methods to estimate GHF

In addition to the few and sparse penetrative GHF estimates in Antarctica, continental (Fig. 5) and regional (Fig.
6) estimates have been derived from both solid Earth (geophysical/geological), and glaciological data and models.

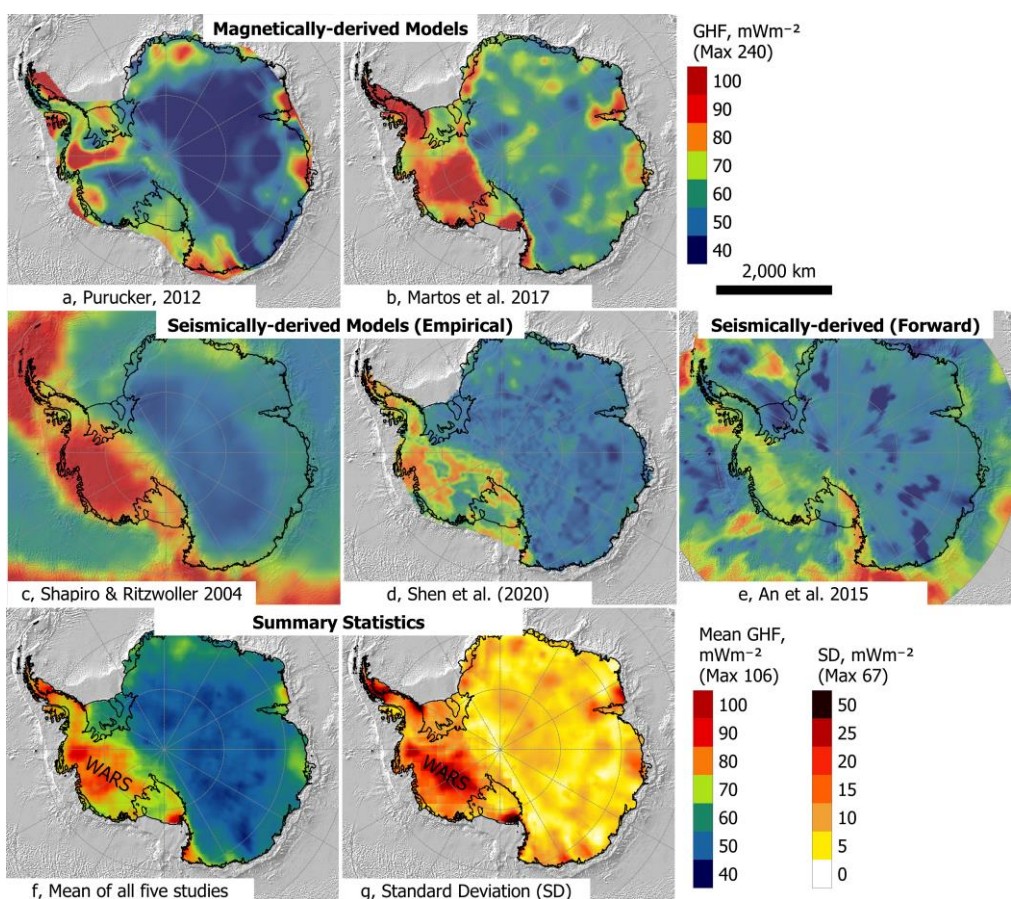

**Fig. 5. Continent-scale geophysical estimates of GHF derived from magnetic Curie depth estimates (a and b; Martos et al., 2017, and Purucker, 2012 - an update of Fox Maule et al, 2005) and seismic models (c to e; An et al., 2015b; Shapiro and Ritzwoller, 2004; Shen et al., 2020). The mean and standard deviation of the combined studies are given in f and g, (available in the Supplementary Material), highlighting the large disparities in West Antarctica. WARS – West Antarctic Rift System.**

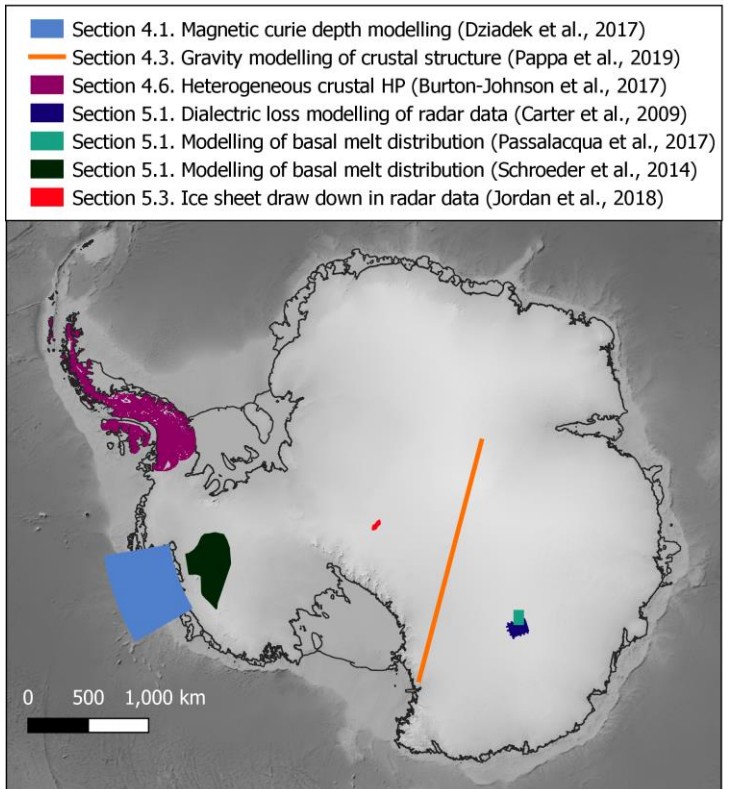

328

**Fig. 6. Coverage of sub-continental scale regional estimates of GHF, with reference to the section where the data is**
**discussed.**

### 4.1. **Magnetic-derived estimates**

As for the penetrative methods of GHF estimation described above (Section 3), geophysical methods also derive
GHF from a temperature gradient. In this case, magnetic survey data is used to determine the depth at which the
maximum temperature of ferromagnetic magnetisation is exceeded (the Curie temperature; Haggerty, 1978). This
Curie temperature is different for different minerals, but is assumed in these studies to be the Curie temperature
of magnetite (580 °C) as this mineral is most commonly the dominant contributor to crustal magnetisation (Bansal
et al., 2011; Fox Maule et al., 2005; Langel and Hinze, 1998).

Above the Curie temperature, rocks lose their ability to maintain ferromagnetic magnetisation (e.g. Haggerty,
1978). The depth of this isotherm in the crust (the Curie Point Depth, CPD; Fig. 7 and Fig. 2) is thus assumed to
be the depth to the bottom of the magnetic source (DBMS) determined from magnetic survey data. The DBMS
maps a transition zone, rather than an exact depth (Haggerty, 1978), and can provide information on crustal
temperatures at depths not accessible by other means (Andrés et al., 2018; Okubo et al., 1985). Regions found to
have a shallower DBMS (and thus an assumed shallower CPD) are expected to have higher average temperature
gradients, and, therefore, higher GHF (e.g. Aboud et al., 2011; Andrés et al., 2018; Arnaiz-Rodríguez and
Orihuela, 2013; Bansal et al., 2013, 2011; Bhattacharyya and Leu, 1975; Guimarães et al., 2013; Li et al., 2017;
Obande et al., 2014; Okubo et al., 1985; Ross et al., 2006; Salem et al., 2014; Tanaka et al., 1999; Trifonova et
al., 2009).

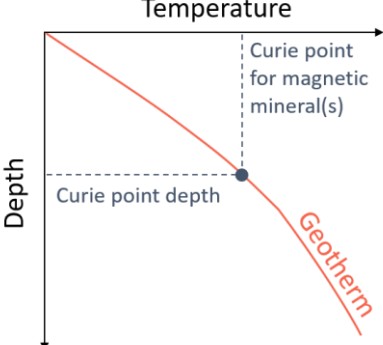

348

**Fig. 7. Approximation of the geothermal gradient from the Curie point depth (CPD). The CPD is assumed to mark the base of the magnetic crust (DBMS).**

The first Antarctic-wide magnetically-derived GHF map (Fox Maule et al., 2005; updated by Purucker, 2012, Fig. 5a) used the "equivalent source magnetic dipole method" (Mayhew, 1979) to map magnetic anomalies from multiple satellites at different altitudes as evenly distributed magnetic dipoles on the Earth's surface (Dyment and Arkani-Hamed, 1998). Due to filtering of the data during processing, this magnetic anomaly distribution is only susceptible to shallow, short-wavelength magnetic variation. To calculate the CPD, a long-wavelength CPD model was modified until it reproduced the determined short-wavelength anomalies. The temperature gradient represented by this CPD was combined with assumed homogenous crustal properties (heat production and conductivity) to model the surface heat flow. Due to the high altitude of the satellite data, the horizontal resolution of this approach was limited to at least a few hundred kilometres.

Spectral methods are the alternative and more commonly applied approach to estimating the DBMS, analysing the spectrum of wavelengths in magnetic profiles or gridded data (e.g. Blakely, 1996; Okubo et al., 1985; Spector and Grant, 1970). These methods depend on the implicit assumption that long wavelength features result from deep sources. The depth of this source is calculated from a "power spectrum" (Fig. 8) of wavenumber (the inverse of the wavelength) against the logarithm of each wavenumber's "power" (the square of each wavelength's magnitude after conversion by a Fast Fourier Transformation to describe the spectrum of wavelengths in the signal). From this power spectrum (Fig. 8) the top ($Z_t$) and centre ($Z_0$) of the deepest magnetic layer are inferred from the slope of the intermediate and long wavelength zone of the spectra derived from magnetic anomaly data. The DBMS ($Z_{DBMS}$) stems from the simple geometric relationship between these depths:

$$Z_{DBMS} = 2Z_0 - Z_t$$

(6)

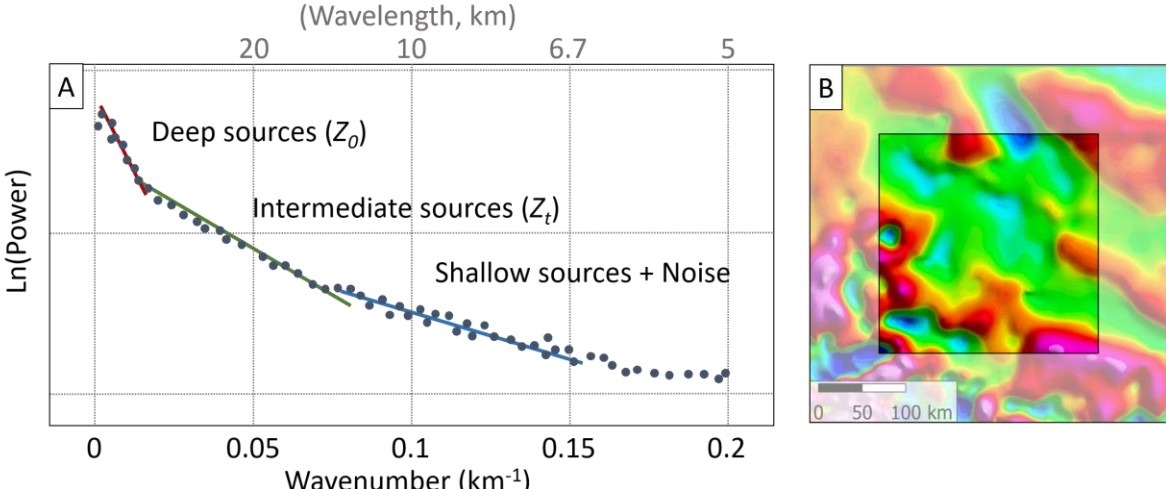

**Fig. 8. A) Identification of the slopes of the intermediate and long wavelength magnetic anomalies from the power spectrum of magnetic anomalies within a single magnetic window (B). For illustration, small circular anomalies in the magnetic window (B) would correspond to shallow sources in the power spectrum, whilst larger anomalies would correspond to intermediate and deep sources.**

To map the DBMS across a study area, the spectra of magnetic anomalies are computed within overlapping rectangular windows regularly spaced over the aeromagnetic map. Particularly for gridded data, the dimensions of the region chosen to analyse the long wavelength frequencies must be sufficiently large to capture the DBMS. Ravat et al. (2007) elaborate that the dimension of the region analysed may need to be (in some cases) up to 10 times the DBMS, but that dimensions exceeding 200 to 300 km may average different large-scale crustal structures. This suggests that satellite data, which typically detects magnetic anomalies in that wavelength, may not be suitable for this spectral method of CPD estimation. Choosing the window size therefore forces a trade-off between accurately determining the DBMS within each sub-region and resolving small changes in DBMS between sub-regions (Ross et al., 2006).

Spectral methods have been applied in Antarctica (Dziadek et al., 2017; Martos et al., 2017a; Purucker and Whaler, 2007; Fig. 5b and Fig. 6) to combined satellite and airborne magnetic anomaly data (e.g. ADMAP; Golynsky et al., 2006; Maus, 2010). The results show a general agreement at a continental scale, but vary significantly on a regional scale (Fig. 5). This is related to the resolution of the magnetic anomaly data, particularly in regions where only satellite magnetic data are available. Furthermore, regional-scale magnetic anomaly databases are usually a mosaic of individual aeromagnetic surveys. Ross et al. (2006) emphasise that subtle discontinuities along survey boundaries are caused by differences in survey specifications, such as flight line spacing, flight altitude, regional field removal, or the quality of data acquisition. These, for instance, may contaminate the long-wavelength signal caused by deep magnetic sources (Grauch, 1993). Long wavelength features can also result from shallow but spatially extensive sources, such as volcanic provinces, and can lead to an underestimation of the DBMS.

CPD estimates assume a homogenous magnetic mineralogy of magnetite, and thus a Curie temperature of 580 °C (Bansal et al., 2011; Fox Maule et al., 2005; Langel and Hinze, 1998). This assumption neglects the compositional variability in plutonic rocks that lead to Curie temperature ranges between 300 °C and 680 °C, and in cases of

magnetic assemblages of Fe-Ni-Co-Cu metal alloys up to 620 °C to 1084 °C (Haggerty, 1978). Without further
constraints and validations, these assumptions remain the best approach, especially in sparsely sampled regions
like Antarctica, but introduce uncertainties of several kilometres in Curie depths and consequent uncertainties in
GHF estimates (Bansal et al., 2011; Ravat et al., 2007). Similarly, in areas of thin crust, non-magnetic mantle
rocks can be shallower than the Curie depth. In these regions, the calculated CPD will appear shallower due to a
lack of magnetic minerals in the mantle rocks (Fig. 9.; Frost and Shive, 1986; Wasilewski and Mayhew, 1992).
This can be investigated through comparison of the Antarctic Curie depth estimates with the seismically- or
gravitationally-derived depth of the crust-mantle boundary (the Moho depth; Fig. 10 and Fig. 2). For example,
thermal modelling of seismic, gravity, and magnetic data showed the DBMS of the Norwegian margin reflected
the basement geometry, not the CPD, and that surface heat flow estimates using magnetic CPD models were thus
unreasonably high (Ebbing et al., 2009).

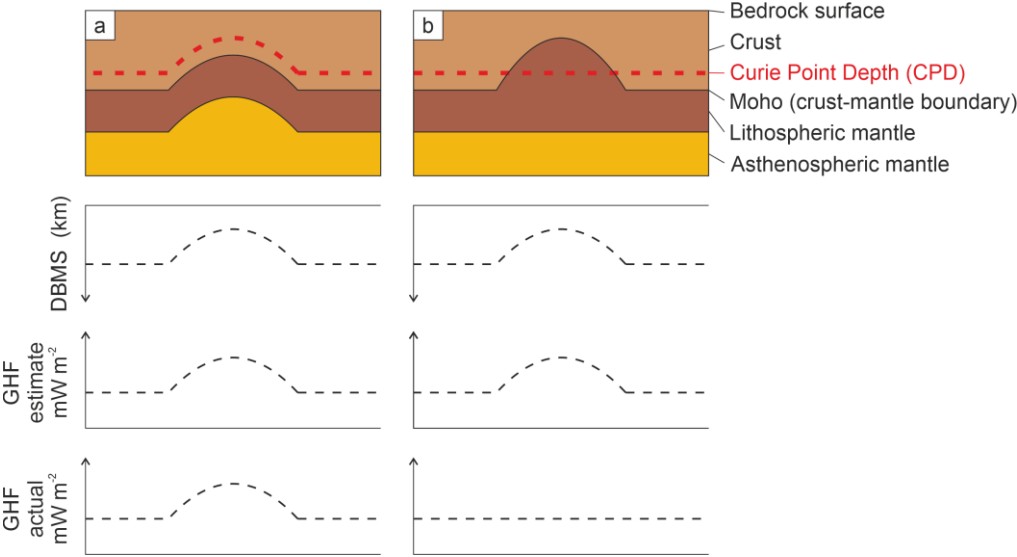


**Fig. 9. Two scenarios illustrating the ambiguity in estimating Curie point depth (CPD) and GHF. a) Estimates from a**
**region with a shallow CPD over an area of thin crust. b) Similar but incorrectly interpreted estimates from a region of**
**shallow non-magnetic mantle rocks. In scenario (b), the DBMS is shallower despite there being no deviation in the CPD**
**depth. DBMS: Depth to the bottom of the magnetic source (assumed to represent the CPD in the GHF estimates**
**discussed).**

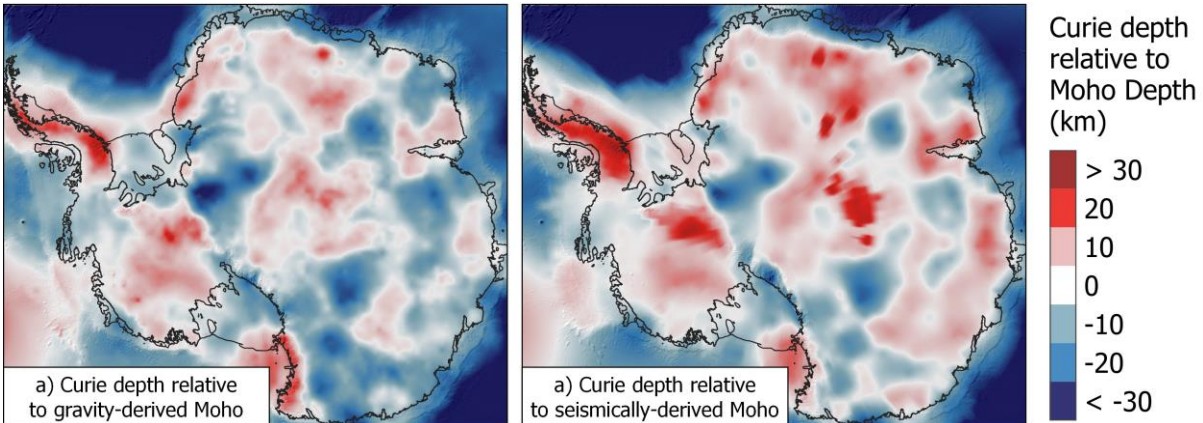


 **Fig. 10. Comparison of Curie depth (Martos et al., 2017) and depth of the crust-mantle boundary (the Moho depth)**
**derived from a) gravity modelling (Pappa et al., 2019b), and b) seismic modelling (An et al., 2015a). Negative values**
**show areas where the estimated Curie depth is deeper than the estimated Moho depth, and positive values are where**
**the Curie depth is shallower than the Moho depth.**

However, whilst in general the Earth's mantle does not contribute to the magnetic signal (due to its weak magnetisation and high temperature conditions), in some cases the Curie depth may indeed lie within the mantle. This occurs where metallic magnetic phases in the mantle beneath old, tectonically stable crust ("cratons"; Ferré et al., 2013) or subduction regions (e.g. Blakely et al., 2005) contribute to mantle magnetisation. In these settings the crust-mantle boundary should not be considered an absolute magnetic boundary (Ferré et al., 2013). This implies that if in a given region the Moho depths are shallower than the deepest magnetic layer, a magnetic mantle at temperatures below the Curie temperature may be considered. However, even in these cases the upper mantle susceptibility will be more than 1-2 magnitudes smaller than the overlying crust. This is not considered in current spectral methods assuming constant susceptibility. Consequently, Curie depth methods yield non-unique solutions, and further available constraints and observations need to be considered, when interpreting the Curie temperature distribution (e.g. geological evidence, borehole measurements, and Moho depth estimates).

4.2. **Seismic-derived estimates**

Temperature is the dominant control on seismic velocity in the mantle (e.g. Carlson et al., 2005), and hence the mantle heat flow at the base of the Antarctic crust can be determined from seismic data. By determining the change in seismic velocities marking the density discontinuity at the lithosphere-asthenosphere boundary (Fig. 2) the depth of the 1330°C isotherm can be estimated. This is the "mantle adiabat" marking the top of the seismic low-velocity zone, and the change from a solid to ductile mantle (Fig. 2). The continental-scale GHF can then be estimated by assuming the heat production and conductivity of the lithosphere above this boundary, and integrating this with the seismically-derived mantle heat flow (An et al., 2015b; Fig. 5d). However, the seismically-derived, continent-scale Antarctic GHF model of An et al. (2015a) (Fig. 5d) is limited to a lateral spatial resolution of >120 km, assumes a laterally uniform crustal structure, and is insensitive to the lithospheric geotherm (instead it inversely correlates with crustal thickness).

Composition also affects seismic velocities. For example, a 2% increase in velocity can be explained either by a 120°C decrease in temperature, a 7.5% depletion in iron, or a 15% depletion in aluminium (Godey et al., 2004). Slow mantle velocities at subduction zones can also be caused by water or hydrous fluids serpentinising the mantle wedge (Fig. 2; Kawakatsu and Watada, 2007). However, velocity in the Antarctic seismic model (An et al., 2015b) does not account for variability of mantle compositions, mineralogy, grain size, or water content of the mantle or crust. An uncertainty in the lithospheric thickness of 15-30 km was assumed by (An et al., 2015b) based on the 150°C temperature uncertainty, but ~50 km uncertainty for ~200 km thick lithosphere may me more accurate (Artemieva, 2011; Godey et al., 2004). In addition, seismological models suffer from limited and inconsistent spatial coverage, which can lead to discrepancies in upper mantle velocities and differences in Moho depths (Fig. 2) up to 10 km, even for the same receiving station (An et al., 2015b supporting information; Pappa et al., 2019).

Some constraints on the mantle and lithosphere composition can be determined from xenoliths (rock fragments of the deep crust or mantle entrained in magma rising from depth) or exposed deep crustal sections, where

variation in temperature and composition with depth can be determined from the metamorphic minerals present. Constraints can also be derived empirically by comparing the seismic velocity with similar regions. Shapiro and Ritzwoller (2004) (Fig. 5c) extrapolated global heat flow measurements to Antarctica based on the assumption that structurally similar regions have similar magnitudes of GHF. This was achieved by calculating a spatially variable "similarity functional" determined from the differences between the seismic velocity and seismic Moho depth between a location of interest and a comparable location elsewhere. A histogram of heat flow measurements could then be assigned to the location of interest in Antarctica based on the similarity-weighted sum of measurements from structurally similar regions, and the mean values of these distributions mapped as continental heat flow. Spatial resolution was limited to the lateral resolution of the global shear velocity model across Antarctica (600-1000 km; Shapiro and Ritzwoller, 2002). Although the studies of Shapiro and Ritzwoller (2004) and An et al. (2015a) both used seismic data and are thus frequently compared, it is important to highlight that they use very different approaches in deriving heat flow (the former employing a probabilistic approach and the latter using forward modelling).

The empirical seismically-derived model for Antarctica has recently been revised (Fig. 5d; Shen et al., 2020). Rather than the low-resolution global database used by Shapiro and Ritzwoller (2004), an Antarctic seismic model was derived and compared with the high-resolution seismic model and GHF measurements of the USA; again calculating spatially variable similarity functionals to compare the data. Recognising the non-unique solutions provided by this method, Shen et al. (2020) also map the associated uncertainties of their model.

### 4.3. Gravity model-derived estimates

Satellite gravity data has been used as an alternative to seismic modelling to determine crustal thickness. Pappa et al. (2019b) used satellite gravity data, a model of global gravity variation (the "geoid"), surface and bedrock topography, and assumed rock and ice densities to calculate the topographically-corrected variation of gravity in Antarctica (the "Bouguer anomaly"), from which the depth of the crust-mantle boundary could be calculated. This approach to calculate crustal thickness is sensitive to long-wavelength (>150 km) features representing deep structures, rather than short-wavelength, near surface density changes. However, gravity-modelling solutions are non-unique, and require additional constraints on the density contrast between the crust and mantle at a reference depth, and/or seismic depth constraints on crustal thickness.

Using the gravity-derived crustal thickness estimates, cross-sectional models of the mantle and lithospheric structure were calculated, with adjustments made to crustal density and crustal thickness until the models reflected the observed variation in gravity and elevation (Pappa et al., 2019b). By assigning assumed values of heat productivity and thermal conductivity values to the modelled cross-sections, surface heat flow was calculated along the line of the modelled cross-section (Fig. 6).

### 4.4. Conjugate margin-derived estimates

An alternative approach to constrain the probable GHF of East Antarctica is to compare it with its Gondwanan conjugate margins, reconstructed prior to the breakup of the supercontinent (Fig. 11). Plate tectonic reconstructions indicate that the subglacial geology of East Antarctica is comparable to the margins of Australia, Africa, and India (Aitken et al., 2016; Daczko et al., 2018; Ferraccioli et al., 2011; Flowerdew et al., 2013; Mulder

et al., 2019). By kriging the heat flow measurements of the continents in their pre-Gondwana breakup arrangement, Pollett et al. (2019) interpolated a heat flow surface through Antarctica and its conjugate margins (Fig. 11). This method highlighted similarities and differences between the most recent seismic and magnetically derived geophysical models of Antarctic heat flow (An et al., 2015b; Martos et al., 2017) with the better constrained heat flow of the conjugate margins. In particular, this approach showed reasonable agreement along the margin with Africa, but an absence in either the magnetic or seismic models of high heat flow provinces in East Antarctica comparable with south Australia; an absence of the low heat flow of SW Australia in the magnetically-derived model of East Antarctica (Martos et al., 2017); and an absence of the high heat flow of northern India in the seismically-derived model of East Antarctica (An et al., 2015b). However, when extrapolating heat flow away from the conjugate margins into the interior of Antarctica, this approach is susceptible to the method of interpolation used and the quality and scarcity of the borehole-derived GHF estimates in the interior of Antarctica (Section 3).

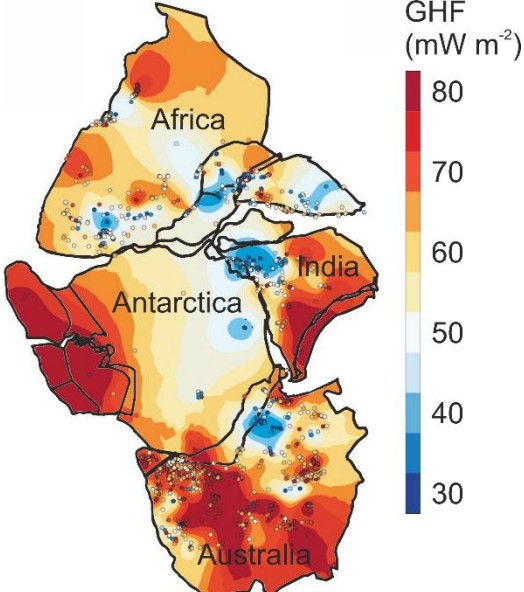

**Fig. 11. Interpolated heat flow map of Gondwana, showing the derivation of Antarctic GHF from the reconstructed conjugate margins of the supercontinent. Terrestrial heat flow data shown by points. Adapted from Pollett et al. (2019).**

### 4.5. Isostatic elevation

In addition to crustal thickness and density, the thermal state of the lithosphere also contributes to its isostasy and observed surface elevation. The effect of thermal isostasy on the bathymetry of oceanic crust is well recognised: as oceanic crust migrates from the spreading ridge it cools, thickens, contracts, and subsides (Stein and Stein, 1992). However, the effect of thermal isostasy on continents is masked by compositional contributions to isostatic elevation (i.e. lateral variations in crustal thickness and density, Fig. 12a; Hasterok and Chapman, 2007b, 2007a).

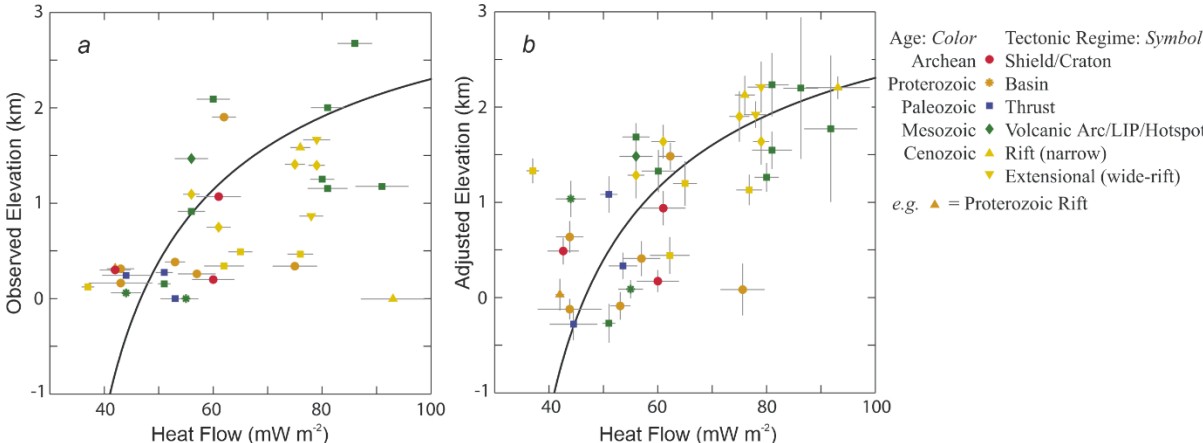

**Fig. 12. Relationship of the median observed (a) and adjusted (b) elevation and median compiled heat flow values of 36 geological provinces on the land and continental shelves of North America, ranging from 30 - 2082 x 10³ km². Compiled heat flow data excluded values outside of the range 20 - 120 mW m⁻² as these values were most likely affected by near surface processes (e.g. hydrothermal circulation) or shallow magmatism, and do not reflect the lithosphere's thermal state. Observed elevations are converted to adjusted elevation by normalising according to their seismically-derived crustal thickness and crustal density and an equation for thickness and density-based isostasy. The black curve shows the best-fitting thermal-isostatic model for North American adjusted elevation and heat flow. Adapted from Hasterok and Chapman (2007a).**

Hasterok and Chapman (2007b, 2007a) developed a methodology for investigating thermal isostasy in the continental lithosphere by normalising the observed elevation using an isostatic correction. The calculated compositionally-corrected elevation generally increases with increasing surface heat flow (Fig. 12b). This approach was used to derive the thermal contribution to isostatic elevation of Australia and North America, and estimate the continental sub-lithospheric and radiogenic heat flow (Hasterok and Chapman, 2007b; Hasterok and Gard, 2016). Whilst in general, the compositionally-corrected elevation and surface heat flow values followed the modelled curve for thermal isostatic equilibrium (Fig. 12b), anomalous regions lie away from this curve. These anomalies result from: 1) additional sources of buoyancy and/or dynamic support (e.g. anomalously buoyant mantle lithosphere); 2) anomalous surface heat flow, not representative of the deeper thermal regime (e.g. high concentration of heat producing elements in the shallow crust); 3) deviations from the thermal properties of the reference crustal model (e.g. heat production); or 4) combinations of these properties (Hasterok and Gard, 2016).

Although developed for regions of known heat flow, application of this approach to Antarctica (Hasterok et al., 2019) may provide an alternative estimate of heat flow based largely on two well-constrained variables: surface and bedrock topography. However, it is dependent on the quality of constraints on crustal thickness, density, heat production, and thermophysical properties of the upper crust (of which uncertainty in upper crustal heat production has the largest effect; Hasterok and Chapman, 2007b). For example, regions where high surface heat flow is dominantly from anomalously high upper crustal heat production will have lower elevations than regions of similar surface heat flow but with lower upper crustal heat production. Crust that has experienced tectonic and magmatic activity in the Cenozoic (i.e. <66 Ma) may be in a transient rather steady-state thermal regime, so this approach may have challenges in West Antarctica. Steady-state thermal modelling is thus more applicable to the old, stable crust of East Antarctica; particularly if the heat flow and isostasy of the conjugate margins are considered (Hasterok and Gard, 2016; Pollett et al., 2019). However, differences between the crustal thickness

based on gravity modelling and isostatic elevation modelling may indicate variable densities and/or compositions
of the underlying mantle (Pappa et al., 2019b, 2019a).
4.6. **Enhancement of GHF estimates by incorporation of heterogeneous crustal compositions**
The geophysical approaches described above assume laterally homogenous heat production in the crust. However,
given the geologically heterogeneous composition of the crust, it is important to consider the effects of variable
lithospheric heat production and incorporate this into forward models of GHF.
Radiogenic heat production in the upper crust contributes an estimated 26-40 % of the total continental GHF
(Artemieva and Mooney, 2001; Hasterok and Chapman, 2007b, 2011; Pollack and Chapman, 1977; Vitorello and
Pollack, 1980). Radioactive isotopes of the heat producing elements (HPEs) uranium, thorium, and potassium (U,
Th, and K) are responsible for ~98% of lithospheric heat production (Beardsmore and Cull, 2001). These elements
are incompatible with mineral structures in the mantle and lower crust, so concentrate in the upper crust and
decrease in abundance with depth during planetary differentiation (the chemical and physical separation of an
initially homogenous planetary body into one with an iron-rich core, magnesium-silicate-rich mantle, and a thin
silicate-rich crust; Roy et al., 1968; Rudnick and Fountain, 1995).
The upper crust itself is highly heterogeneous in composition. HPE distribution is determined by their
compatibility in different minerals, concentrating them in Si-rich silicic rocks (e.g. granite or rhyolite) relative to
Fe-rich mafic rocks (e.g. gabbro or basalt). Immature sediments inherit the HPE abundance of their eroded source
rocks, but decrease in HPE abundance with increasing maturity and the consequent decrease in their lithic contents
(Burton-Johnson et al., 2017; Rybach, 1986). Crustal heat production is thus heterogeneous, and the most
significant control of HPE abundance and resultant heat production in the lithosphere is the distribution of the
composite lithologies of the upper crust (Lachenbruch, 1968; Sandiford and McLaren, 2002; Taylor and
McLennan, 1985).
4.6.1. **Whole rock geochemical analysis of heat production**
Heat production of exposed lithologies can be determined from their concentrations of HPE (U, Th, and K)
determined by geochemical analysis, or by airborne or ground-based gamma ray surveys. Radiogenic heat
production for each sample ($H$, μWm$^{-3}$) for the present day ($t=0$) can be determined from Equation 7 (Turcotte
and Schubert, 2014):
$$H = (0.9928 C_0^U H^{U238} + 0.0071 C_0^U H^{U235} + C_0^{Th} H^{Th232} + 0.000119 C_0^K H^{K40})D$$

572 (7)

Where $C_0^U$, $C_0^{Th}$ and $C_0^K$ are the measured concentrations (ppm) of U, Th and K respectively; $H^{U238}$, $H^{U235}$, $H^{Th232}$
and $H^{K40}$ are the heat productivities of the respective isotopes $^{238}$U (9.37x10$^{-5}$ Wkg$^{-1}$), $^{235}$U (5.69x10$^{-4}$ Wkg$^{-1}$),
$^{232}$Th (2.69x10$^{-5}$ Wkg$^{-1}$) and $^{40}$K (2.79x10$^{-5}$ Wkg$^{-1}$); and $D$ is the assumed density of the rock (e.g. 2800, 2850,
and 3000 kg m$^{-3}$ for felsic, intermediate, and mafic granulites, respectively; Hasterok and Chapman, 2011). When
using geochemical data to calculate heat production, this allows new and archive data to be used to calculate the
heat production of the sampled outcrop. However, many archive analyses occurred prior to the development of
accurate U quantification (e.g. by high resolution XRF or ICP-MS). An empirical relationship (Equation 8;
Burton-Johnson et al., 2017) allows calculation of total U, Th, and K heat production ($H$) from samples possessing
only Th and K data ($H_{K,Th}$; correlation coefficient, $R^2 = 0.9$; Fig. 13).
$$H = 1.4H_{K,Th} + 0.3$$

583  (8)

Heat production values can be assigned to bedrock geology either by interpolation of the point values or by
assigning the point values to the mapped geology and assigning their average value to the geological unit; the
average being either the mean (Veikkolainen and Kukkonen, 2019), area weighted mean (Slagstad, 2008), or
median value (Burton-Johnson et al., 2017). Interpolation shows spatial variability within a unit, but is affected
by the interpolation method used, requires sufficient and evenly distributed data coverage, and is affected by
anomalous values. For these reasons, the median values were used for the unevenly distributed archive data of the
Antarctic Peninsula (Burton-Johnson et al., 2017). In Antarctica, maps of median (Antarctic Peninsula, Fig. 6;
Burton-Johnson et al., 2017) and transects of mean (coastal East Antarctica; Carson et al., 2014; Carson and
Pittard, 2012) heat production data have been integrated with geophysical models of the deeper heat flow to
estimate the total GHF at the bedrock surface.
Integrating spatially variable upper crustal heat production into the geophysical models of Antarctic GHF resulted
in increased estimated spatial GHF variability, including local regions of high GHF above HPE-enriched granitic
intrusions (Carson et al., 2014; Leat et al., 2018). The relative concentration of the HPE into the upper crust may
result in it contributing a highly variable 6-70% of the total GHF, although 3D crustal modelling is required to
constrain its thickness (Burton-Johnson et al., 2017). This modelling also showed the impact of sedimentary basins
on GHF distribution, as thick, extensive units of immature, clay-rich sediments may form extensive regions of
enhanced GHF, even though more mature, quartz-rich sediments are associated with low GHF (Burton-Johnson
et al., 2017). This highlights the importance of accurately constraining the upper crustal geology and its chemistry
when estimating GHF from geophysical data.

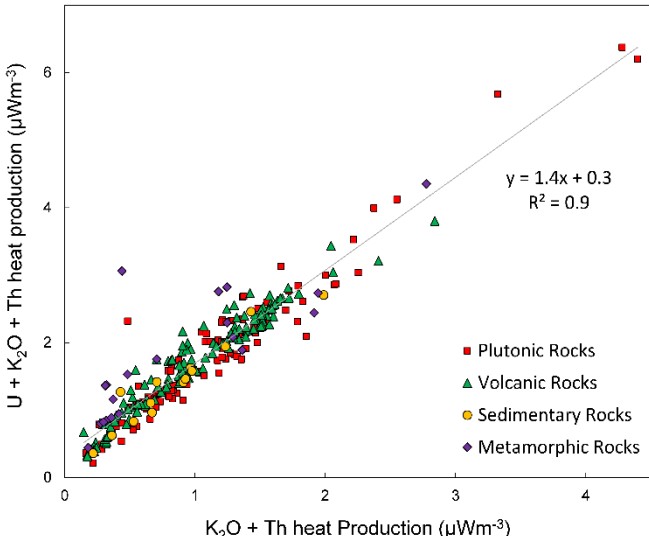


Fig. 13. The relationship between total calculated heat production from U, $K_2O$ and Th decay and the heat production values from $K_2O$ and Th only for different broad lithologies, enabling total heat production calculation from incomplete archive data (n = 319; Burton-Johnson et al., 2017).

### 4.6.2. Glacially-derived rock clasts

Although heat production can be determined for exposed bedrock, the likely heat production of the rocks beneath the Antarctic ice sheet is harder to constrain. To investigate East Antarctica, glacial clasts were sampled from moraines adjacent to the Transantarctic Mountains (Goodge, 2018). Granitic samples older than 500 Ma (Ross Orogen) were selected as likely lithologies of the interior of East Antarctica, as these are the dominant lithologies of other Precambrian cratons (>542 Ma regions of tectonically-stable continental crust; e.g. central Canada). These clasts were analysed for their HPE abundance and attributed to their likely source area (the drainage basin of their associated glaciers). A probable range of subglacial heat flow values was estimated by assuming mantle and lower crustal GHF values and a thickness for the upper crust based on other Precambrian shields. This indicates that East Antarctic heat flow is comparable to other Precambrian cratons, and comparable to geophysical models of East Antarctic heat flow (Van Liefferinge and Pattyn, 2013). However, broader application of this approach is biased towards more erosion resistant rock types, whilst less competent lithologies will not be preserved after glacial transport and deposition.

### 4.6.3. Gamma ray spectrometry

Rather than whole rock geochemical analysis, the gamma ray spectrum can be used to determine the concentrations of radioactive isotopes, including those of K, Th, and U, and was first used for U exploration. Gamma ray spectrometry can be surveyed in the field, on samples, or from the air. Airborne surveys can cover large areas, and have been used to survey Western Australia, SW England, and all of Finland (Beamish and Busby, 2016; Bodorkos et al., 2004; Hyvönen et al., 1972). However, the data requires multiple corrections, and the recorded data integrates the radiation from the bedrock, surface cover (including soil and vegetation), the atmosphere, cosmic radiation, and the aircraft, making the data less accurate than ground measurements or sample analysis (Veikkolainen and Kukkonen, 2019). The technique is only sensitive to the upper 25cm of the land surface, with overlying sediments and water bodies masking the radiation and leading to underestimates of heat production (Phaneuf and Mareschal, 2014). However, if the signal could be linked to mapped geological units and other evidence for subglacial geology (e.g. aeromagnetic and gravity anomalies) it may be feasible to extrapolate the calculated heat production beneath the ice sheet. Hand-held gamma ray spectrometry studies, where heat production can be correlated with lithology along exhumed crustal profiles, show promise in this regard elsewhere (Alessio et al., 2018).

### 4.6.4. Crustal structure

Whilst surface HPE distribution can be constrained by measurements, the vertical distribution is more ambiguous. In heat flow models, heat production is often assumed to decrease exponentially with depth (e.g. Fox Maule et al., 2005; Martos et al., 2017). This exponential model was developed to explain observations from exposures of large, thick composite granite bodies (batholiths) where magma was initially emplaced at different depths in the crust (Lachenbruch, 1968, 1970; Swanberg, 1972) and reflects a proposed decrease in HPE abundance with

increasing metamorphic grade (Lachenbruch, 1968; Sandiford and McLaren, 2002). However, this relationship
has been challenged by other studies comparing HPE abundance and metamorphic grade (Alessio et al., 2018;
Veikkolainen and Kukkonen, 2019), showing that the lithological change from the largely silicic upper crust to
the mafic lower crust has a larger influence on HPE abundance than metamorphic grade (Bea, 2012; Bea and
Montero, 1999). Deep (9-12 km) boreholes also show a correlation of heat production with lithology, but not with
depth (Clauser et al., 1997; Popov et al., 1999). In fact, heat production *increased* for the first 2 km of the 12 km
superdeep well of the Kola Peninsula, Russia, then remained variable but high with increasing depth (Popov et
al., 1999). Similarly, heat production increases below 3 km in the recent 5 km UD-1 well of the Cornubian
Batholith, UK (Dalby et al., 2020). As such, the available evidence indicates that the first-order HPE distribution
is controlled by the HPE abundance of the crust prior to metamorphism and the vertical distribution of the crust's
composite rock types. Inversely, it indicates that HPE distribution is not controlled by depth in the crust or the
degree of metamorphism resulting from the increase in pressure and temperature.
Without evidence for the deeper structure of the crustal column, the lithological and HPE distribution of the
lithosphere can instead be modelled as layers of variable thickness and heat production: the upper crust, middle
crust, lower crust, and mantle lithosphere. Surface heat flow is largely insensitive to variations in the heat
production or thickness of the mafic lower crust and mantle lithosphere due to their heat production being ~1-2
orders of magnitude lower than that of the upper crust (Hasterok and Chapman, 2011; Rudnick and Fountain,
1995; Rudnick et al., 1998). The middle crustal layer can either be excluded (Hasterok and Chapman, 2011) or
treated as a layer of invariable heat production (e.g. An et al., 2015, for Antarctica) due to its low heat production
compared with the range of the upper crust. Lithospheric heat production can thus be defined by the heat
production and relative thickness of the upper crust, or upper crustal heat producing layer (Hasterok and Chapman,
2011). This can be defined by:
$$Q_s = Q_b + H_{UC}D = FQ_s + H_{UC}D = H_{UC}D/(1 - F)$$

664   (9)

Where $Q_s$ is the surface heat flow, $Q_b$ is the basal heat flow of the upper crust, $H_{UC}$ is upper crustal heat production,
$D$ is the thickness of the upper crustal heat producing layer, and $F$ is the proportion of the surface heat flow
contributed by the basal heat flow ($Q_b$) (adapted from Hasterok and Chapman, 2011).
Rather than a simple layered model, more complex 2D or 3D models of upper crustal structure can be developed
using geophysical data, and the 2D or 3D crustal units assigned heat production and conductivity values based on
analyses of representative exposures. A 3D crustal model derived from gravity and aeromagnetic data was
developed to map heat flow in Norway (Ebbing et al., 2006; Olesen et al., 2007). In Antarctica, this has been
applied in 2D to the high heat production granites of the Ellsworth-Whitmore Mountains using airborne magnetic
and gravity data and bedrock topography (Leat et al., 2018), and the Transantarctic Mountains using topography
and satellite gravity data (Pappa et al., 2019b).
Even though variability in deep lithospheric heat production has a smaller effect on surface heat flow than
variability in upper crustal heat production (Hasterok and Chapman, 2011), it is not homogenous. These
thermophysical properties can be constrained from deep xenoliths (fragments of rock entrained in magma rising
from depth) (Hasterok and Chapman, 2011; Martin et al., 2014) and crustal sections (Berg et al., 1989), which
can also inform on the local geothermal gradient at the time of their crystallisation.
To help constrain the properties of the Antarctic mantle, including its influence on Antarctic heat flow, a
Geological Society of London Memoir is currently being compiled summarising the data gained from mantle
xenoliths (Martin and van der Wal, in prep.). This includes a sample database, and a compilation of their grain
size and water content. These xenoliths are from shallow sources, as their occurrence is biased towards areas of
crustal rifting where the lithosphere is thinner, although some xenoliths are from deeper sources (e.g. from the
Amery Rift and Ferrar Dolerite).

## 5. Glaciological inverse estimation of GHF

Although geothermal heat flow has a geological derivation, it can also be constrained by multiple approaches
through its observable effects on the overlying ice sheet. Inverse modelling can be applied to observed
glaciological properties (e.g. glacial flow and melt rates) and the required GHF calculated. We will describe in
this section different methods used in glaciology to derive GHF.

### 5.1. Subglacial water

The presence of subglacial water can be detected with an ice-penetrating radar. The reflective properties of the
ice-bedrock interface depend on the presence of water and, with certain caveats, radar surveys can be used to map
subglacial water. In general terms, a glaciological model can then be used to estimate the values of GHF that
better predict where basal temperatures reach the pressure melting point and melting occurs. We will describe in
this Section examples of this approach.
Carter et al. (2009) modelled the dielectric loss of radar data through the ice column around Dome C in East
Antarctica (Fig. 6) to infer the basal reflectivity and verify the presence of subglacial water. Because the
temperature profile of the ice sheet is one parameter affecting dielectric loss, this approach required inference of
the basal heat flow from temperature-depth modelling over the last 254 ka. The Shapiro and Ritzwoller (2004)
GHF model was used initially (see section "4.2. Seismic estimates"), but when the calculated vertical ice velocity
($m_W$) at the bed exceeded the initial melt rate ($m_T$), the GHF was modified until $m_T$ and $m_w$ were equal. This
approach identified localised high GHF anomalies, but (excepting these anomalies) they calculated that 66 % of
the study area was either at or near the pressure melting point (anywhere that ice is thicker than 3500 m) without
invoking enhanced GHF.
Schroeder et al. (2014) modelled the spatial distribution of melt beneath the ice sheet in the Thwaites Glacier
catchment (Fig. 6) by mapping the relative bed echo strength of radar data in the region and modelling the water
routing required to match these observations by routing alone (without heterogeneous basal melting). These
routing models were based on the radar-derived ice thickness and surface slope. The 50 selected routing models
were used to model the relative melt required to reproduce the observed echo strengths of each routing model.
This relative melt model was in turn scaled to match the total melt water produced in an ice sheet model of the
Thwaites Glacier incorporating frictional melting, horizontal advection, and an assumed uniform GHF. By
subtracting the frictional and advective contributions, the GHF required to produce the remaining melt could be
calculated. This approach predicted very high heat flow in this region (114 to >200 mW m⁻²), with the highest
heat flow focused around observed and inferred subglacial volcanoes.
With the aim of determining appropriate sites of low basal melting for old-ice drilling, Passalacqua et al. (2017)
also used radar evidence for basal melting and ice sheet modelling to determine GHF around Dome C (Fig. 6).
Wet and dry bed conditions were identified from radar data and ten spots were identified on bedrock topographic
features marking the critical ice thickness where present basal melting becomes possible. These spots were defined
as locations where the upper slopes of the bedrock topography are dry and their lee slopes are wet, with melting
initiating between the two when the ice thickness passes the pressure melting point (Fig. 14). Assuming that GHF
is locally homogeneous between the two bedrock elevations, heat flow was determined by increasing its value in
a 1-D heat model of the local ice thickness until basal melting occurred. These point estimates were interpolated
to generate an approximate map of regional heat flow and calculate basal melt rates over the last 400 ka.

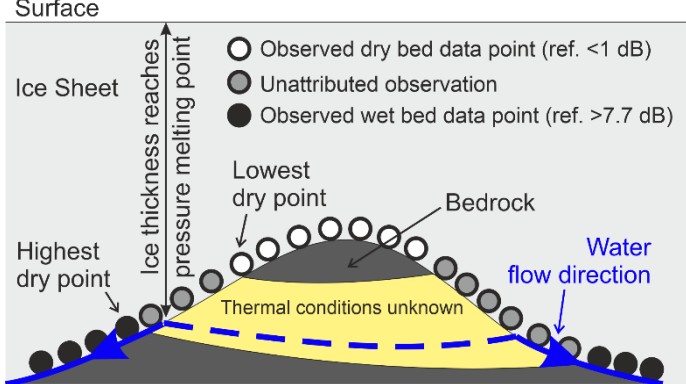


**Fig. 14. Illustration of how the ice thickness exceeding the pressure melting point (PMP) can be identified from radar**
**reflectivity data points, indicating the presence or absence of basal water beneath the ice sheet. Once the PMP is**
**identified, thermal modelling can estimate the required local GHF. Between the thresholds of radar reflectivities**
**representative of wet and dry basal conditions, the thermal conditions are unknown (yellow-shaded region of the**
**bedrock). Adapted from Passalacqua et al. (2017).**
Van Liefferinge and Pattyn (2013) and Van Liefferinge et al. (2018) used steady state and transient
thermodynamic modelling of the East Antarctic Ice Sheet to map the minimum heat flow required to raise the
basal temperature above pressure melting point and generate basal melting. Whilst this was executed to identify
possible sites for drilling the oldest ice in areas that are unlikely to have undergone basal melting in the last 1.5
Ma and did not produce an estimate of absolute GHF, if this approach were combined with other evidence for
basal conditions above the pressure melting point (e.g. combining thermodynamic modelling with subglacial lake
locations), points of minimum heat flow could be mapped.
5.2. **Subglacial lakes**
If temperatures are sufficient for basal melting, and topography depressions are suitable, subglacial lakes can
develop. Subglacial lakes exhibit radio reflectivities 10-20 dB greater than the ice-bedrock boundary, allowing
the current identification of at least 402 lakes beneath the Antarctic ice sheet (Wright and Siegert, 2012).

Whether basal temperatures are sufficient for basal melting and preservation of subglacial lakes is dependent on ice thickness, the surface temperature and accumulation rate, heat transported through ice advection, heat produced by internal deformation and basal sliding, and the GHF. When subglacial lakes are located near ice divides, heat derived by horizontal advection, basal friction, and internal deformation is assumed to be minimal, and thus the heat required to bring the base of the ice sheet above the pressure melting point is a product of ice thickness and GHF. Thus, when subglacial lakes are located near ice divides and the accumulation rate is known (high accumulation rates cool the ice mass), point estimates of *minimum* GHF can be calculated from one-dimensional thermal models of the ice sheet temperature profile, but an assumption that water was derived locally and not routed from elsewhere must also be considered as lakes can only form in topographic depressions. The absence of a lake or basal water does not imply the bed is frozen if the water can drain away (Pattyn, 2010; Siegert and Dowdeswell, 1996).

Conversely, and with the same caveats regarding basal topography and drainage, where the ice sheet is known to be frozen to the bed, the *maximum* GHF can be estimated. For example, Fudge et al. (2019) used the presence of Raymond Arches to deduce where the ice was frozen to the bed at the Siple Coast ice rises to estimate maximum GHF values. Combined, maximum and minimum estimates are more useful than either alone.

### 5.3. **Englacial stratigraphy**

Jordan et al. (2018) identified draw down of internal ice sheet layers and increased bed reflectivity from radar data ~200 km from the South Pole (Fig. 6), indicating enhanced basal melting. Melt rates were calculated using dated radar layers, traced from the Dome C ice-core site, and a depth age model that simulates the draw-down effect of ice from subglacial melt rate. The low ice velocity ($<1.5$ m a$^{-1}$) indicated minimal frictional contribution to basal temperature, and a location at the top of a hydraulic catchment area indicated a low heat contribution from subglacial water. By negating these contributions to heat flow, assuming the basal temperature is at the pressure melting point (and thus could be derived from the ice thickness) and that temporal temperature variations match those of the Dome C ice core, a time-dependent heat equation was applied to the ice sheet to derive the basal GHF required to generate the enhanced melt rates.

### 5.4. **Microwave emissivity**

Englacial temperature profiles have been derived from satellite and airborne passive detection of high frequency L-band microwave radiation (~1.4 GHz; Macelloni et al., 2019, 2016; Passalacqua et al., 2018); data primarily collected to investigate soil moisture and ocean salinity (Kerr et al., 2010). These wavelengths have very low absorption in ice and low scattering by particles (e.g. grainsize and ice bubbles), providing high penetration depths in dry ice.

Macelloni et al. (2019) derived englacial temperature profiles for the Antarctic ice sheet from 2-year averaged vertical-polarised (V) radiation collected at the "Brewster angle" (57.1° ±2.6°; the angle of incidence at which the radiation is perfectly transmitted through the air-snow interface with no reflection, minimising the influence of surface or shallow sub-surface effects). The corrected intensity (brightness temperature, $T_B$) correlates with the surface temperature of the ice, but is also affected by the ice sheet thickness (a largely inverse correlation), density profile, and grain size (Macelloni et al., 2016). As such, the ice sheet's thermal structure at depth could be

estimated by comparing the observed $T_B$ and a simulated $T_B$ derived through microwave emissivity modelling,
including one-dimensional modelling of the ice sheet's temperature profile. Included in the assumed values for
this modelling are the GHF and the accumulation rate; the sources of greatest uncertainty. This method only
applies in areas of slow flowing ice (<10 m yr$^{-1}$), and is optimal in areas of very slow flowing ice (<5 m yr$^{-1}$) as
this negates heating by horizontal ice advection and deformation-derived heat production. It is also only applicable
to areas of thick ice (>1000 m) as the simulations used to model microwave emission do not include bedrock
reflections. This is not a limitation for application to Antarctic GHF research, as it is under these conditions that
heat flow has the greatest influence on ice sheet dynamics.
Comparison of the microwave-derived temperature profile and that simulated by glaciological modelling (Van
Liefferinge and Pattyn, 2013) show good agreement in the upper third of the ice sheet, but diverge in their
temperature estimates with depth, with the largest uncertainties close to the bedrock. This is largely due to
uncertainty in the GHF, but also reflects a decrease in sensitivity of the simulated $T_B$ to the temperature profile
below 1000-1500 m (the bottom 1000-1500 m of the ice sheet contributes <10 % to the total emission). Longer
wavelength emissions (0.5 GHz) with greater sensitivity to the deeper temperature profile may provide greater
accuracy at depth (Jezek et al., 2014). Deep measurements of the ice sheet's temperature profile are required to
validate this method compared to the glaciological models. Although currently limited by its sensitivity to
temperature at depth and the accuracy of the assumed parameters (notably accumulation rate), this approach has
the potential to constrain basal heat flow though variation of the assumed GHF values used in the emissivity
modelling.

## 6. Existing data

Although subglacial borehole-derived estimates of terrestrial GHF are lacking in Antarctica, estimates have been
made from probes into marine sediments and boreholes into exposed bedrock. We have compiled 431 of these
point estimates (Fig. 15; data available in the Supplementary Material and from
https://github.com/RicardaDziadek/Antarctic-GHF-DB). The compiled data originates from multiple methods,
and is variable in its accuracy and limitations, and so we have attempted to qualitatively grade the likely reliability
of each estimate based on specific parameters (Supplementary Material). We do not include values for marine
measurements compiled in the database "Global Heat Flow Data – Abbott Compilation". This database is
available via GeoMapApp and completely undocumented. The labels may point to cruise reports, but not
published data and the data quality remains impossible to evaluate up to this point.

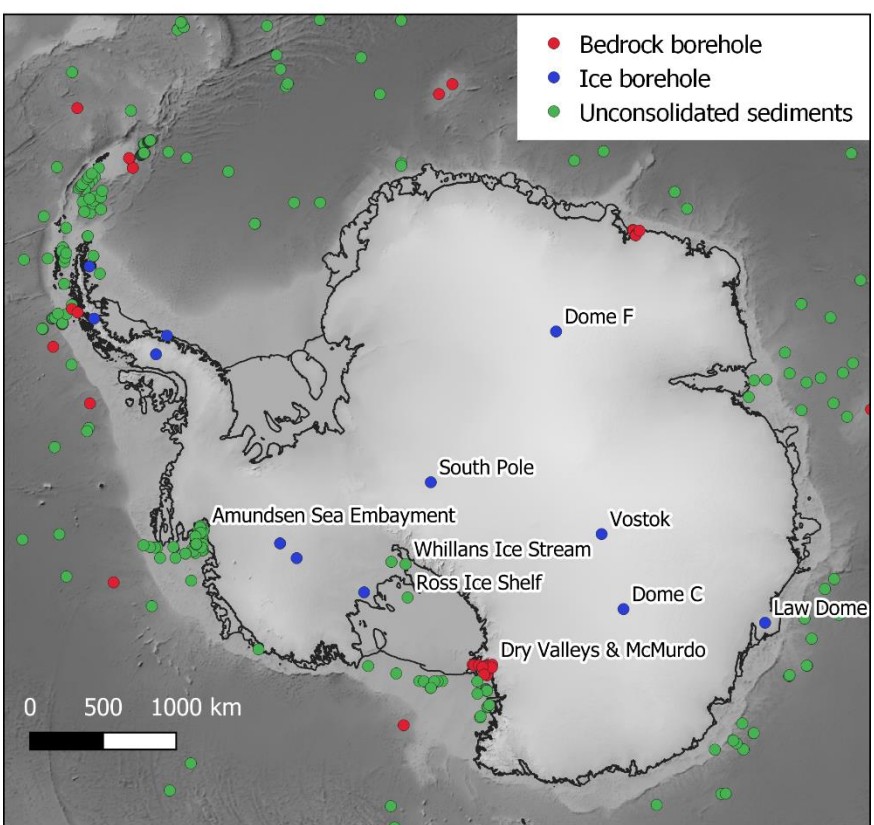

**Fig. 15. Locations of all compiled point estimates of GHF. Database available in the Supplementary Material and from https://github.com/RicardaDziadek/Antarctic-GHF-DB.**

### 6.1. Boreholes into bedrock

Terrestrial, borehole-derived measurements of the geothermal gradient (12 boreholes, Supplementary Material) are limited to the Dry Valleys and McMurdo Sound region (Fig. 15; Bucher, 1980; Decker, 1974; Decker and Bucher, 1982; Pruss et al., 1974; Talalay and Pyne, 2017), and no subglacial terrestrial borehole measurements have been made into the Antarctic bedrock. However, as discussed in Section 3.1., temperature gradients in bedrock must be taken to a sufficient depth to be representative of upward conduction of the GHF rather than downward conduction of the surface temperature. Whilst the GHF estimates from the Dry Valleys Drilling Project (DVDP, including McMurdo Station) were taken from the 75 to >300 m deep boreholes (Bucher, 1980; Decker and Bucher, 1982; Talalay and Pyne, 2017), the shallow 7.6 m borehole from McMurdo Station produces a much higher GHF estimate (164 mW m$^{-2}$, Risk and Hochstein, 1974). This shallow measurement should thus be neglected in preference for the 66 mW m$^{-2}$ value from the 260 m deep DVDP borehole (Decker and Bucher, 1982).

Boreholes into submarine bedrock (34 boreholes, Supplementary Material) have been drilled and temperature gradients measured beneath the McMurdo Sound, Amundsen Sea Embayment, and Ross Ice Shelf (Fig. 15; Bücker et al., 2001; Decker et al., 1975; Gohl et al., 2019; McKay et al., 2018; Morin et al., 2010).

The US Rapid Access Ice Drill project (RAID) aims to achieve the first subglacial, borehole-derived thermal measurements of bedrock following drilling of the overlying ice sheet and coring of ≥25 m of bedrock (Goodge and Severinghaus, 2016).

6.2. **Ice boreholes**
GHF estimates from ice boreholes (15 boreholes, Supplementary Material) are better distributed across the
Antarctic continent than terrestrial bedrock boreholes (Fig. 15). However, not all ice boreholes drilled have been
sufficiently deep or in appropriate sites for GHF estimation (i.e. the ice sheet needs to be stationary and frozen to
the bed). This limits the available GHF estimates to Vostok (Salamatin et al., 1998), Law Dome (Dahl-Jensen et
al., 1999), South Pole (Price et al., 2002), Marie Byrd Land (Clow et al., 2012; Engelhardt, 2004; Gow et al.,
1968), and the Antarctic Peninsula (Mulvaney et al., 2012; Nicholls and Paren, 1993; Zagorodnov et al., 2012)
(Fig. 15). Dome Fuji (Hondoh et al., 2002) is not frozen to the bed, but provides a minimum GHF estimate.
6.3. **Marine and onshore unconsolidated sediments**
The most abundant resource of heat flow estimates from measured temperature profiles around Antarctica comes
from unconsolidated marine sediments (Fig. 15; 362 measurements south of -72° S, Supplementary Material).
However, the data distribution is sparse and heterogeneous, and whilst some regions are well sampled (e.g. the
Amundsen Sea embayment; Dziadek et al., 2019, 2017), other regions (e.g. the Weddell Sea) remain poorly
constrained (Fig. 15). In addition to the open water measurements, two shallow probes (deepest sensors at 1.4 and
0.8 m below the upper sediment surface) have measured the temperature gradient in subglacial sediments below
the Whillans Ice Stream (Begeman et al., 2017; Fisher et al., 2015; see section 3.3.). Two temperature gradients
have also been measured beneath the Ross Ice Shelf (Foster, 1978; Morin et al., 2010), but otherwise heat flow
beneath the Antarctic ice shelves remains poorly constrained regions.
As discussed in Section 3.3, when using these estimates it is important to consider whether the shallow (<~5 m)
temperature gradient recorded by the probe is representative of the deeper GHF, or will have been perturbed by
temperature variation in the overlying ice sheet or water column (e.g. Dziadek et al., 2019). Consequently, the
water depth, the temperature profile of the water column, and possible sources of long-term temperature variation
(e.g. variations in deep water circulation and temperature) should be considered when selecting appropriate point
estimates. Similarly, whilst the shallow temperature gradients measured from Subglacial Lake Whillans (Fisher
et al., 2015), and the Whillans Ice Stream grounding zone (Begeman et al., 2017) are presented as subglacial direct
measurements of Antarctic GHF, by the nature of their location within an ice stream they are not in a thermal
steady state, and the temperature profile will have been affected by long term variation from heat advection and
shear heating. These are effects that cannot be evaluated from their very shallow temperature gradient (0.8 and
1.4 m deep), and accordingly these estimates should be used with caution.
**7.  Current challenges and future research directions**
The collated existing data and methodologies presented above highlight our current limitations in determining the
subglacial GHF of Antarctica and allow discussion of future research.
7.1. **Borehole and probe-derived estimates**
The fundamental limitation for GHF estimation in Antarctica is the lack of borehole-derived estimates from
beneath the Antarctic ice sheet. Without these independent, discrete validation points, the more extensive regional
estimates cannot be accurately evaluated. Therefore, the most promising future development will be the ≥25 m
deep bedrock borehole measurements of the Rapid Access Ice Drill project (RAID; Goodge and Severinghaus,
2016). However, (as noted above) local temperature gradients may not be representative of the regional heat flow,
as local geology, hydrothermal circulation, and topography can result in localised GHF variability. In response,
multiple boreholes where the basal ice is frozen to the bedrock are required to categorise the regional variation,
and topographic effects must be considered and accounted for. Topography may have significant effects on GHF
via its effects on heat diffusion pathways to the surface (Bullard, 1938; Lees, 1910) and must be considered and
investigated in GHF estimates at all scales, including those based on local temperature gradient measurements
(i.e. borehole and probe-derived estimates) and more extensive geophysical and glaciological-derived models.
It is also a necessity that thermal modelling of the bedrock temperature profile for the RAID target sites is executed
prior to drilling to constrain the penetration depth of low-frequency time variation of temperature. Whilst the
RAID target bedrock borehole depth of $\geq$25 m is much shallower than the >100 m borehole depth achieved for
exposed bedrock (Section 3.1.), the overlying ice sheet insulates the bedrock temperature profile from short
duration surface temperature variability (temperature variation penetration depth is dependent on the frequency
of the variation and thermal diffusivity of the material; Carslaw and Jaeger, 1959). However, as is considered for
GHF estimates from ice boreholes (Section 3.2.), low-frequency variation in surface temperatures, heat advection,
and shear heating will all affect the subglacial temperature profile. Consequently, low-frequency temperature
variation must be corrected for, and boreholes are best drilled where the ice is stationary and frozen to the bed (as
is applied to ice borehole selection for GHF estimation). By drilling in such sites where glaciological approaches
are most effective for GHF estimation, the RAID data will allow validation of GHF estimates for the various
englacial temperature methods applied to stationary ice at ice divides (Section 5.). These methods include borehole
temperature profiles, subglacial lakes, ice sheet models, and microwave emissivity. It is thus important that the
englacial temperature profile is measured in addition to the bedrock temperature gradient.
Beyond bedrock drilling there is lot to be gained from further ice borehole drilling. Firstly, existing data must be
evaluated to ensure the methodologies of GHF modelling from borehole temperature profiles are consistent and
accurate. Future ice boreholes into stationary ice frozen to the bed has the potential to supplement the existing
borehole and probe-derived GHF estimates, particularly if the proposed methodology for shallow boreholes can
be validated (600 m depth, or the upper 20% of the ice column; Section 3.2.).
## 7.2. **Geophysical GHF estimates**

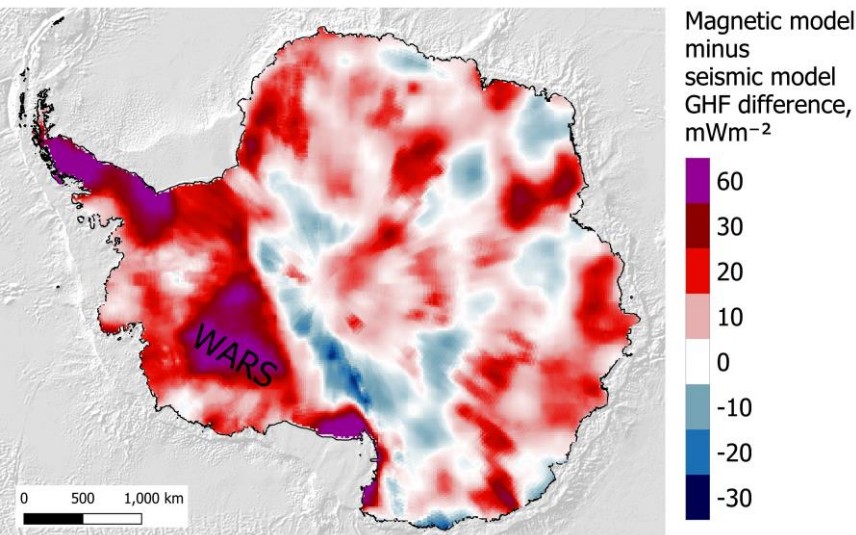

**Fig. 16. Difference in heat flow values between the most recent magnetic (Martos et al., 2017) and forward-modelled**
**seismic (An et al., 2015b) heat flow models. WARS – West Antarctic Rift System.**
Whilst only geophysical methods have provided continental-scale GHF estimates, their values and distribution
vary greatly (Fig. 5, Fig. 16, and Fig. 17). Probability density functions show that whilst there is better agreement
in East Antarctica (Fig. 17a), the seismically derived models estimate more variable and slightly higher GHF than
the magnetically-derived models. In West Antarctica the discrepancies between models are greater (Fig. 5g and
Fig. 17b) even when using similar techniques (compare the empirical seismically-derived estimates of Shapiro
and Ritzwoller, 2002 and Shen et al., 2020, Fig. 17b). However, none of the models of West Antarctica reflects
the GHF distribution of other better-constrained rift systems (Fig. 17c; Lucazeau, 2019), where much more
heterogeneous distributions and a greater proportion of high GHF values (>150 mW m$^{-2}$) are expected.
The fundamental question thus remains: does the West Antarctic Rift System (WARS) have elevated GHF? The
magnetically-derived model of Martos et al. (2017) estimates high GHF, but the most recent forward and
empirically-derived seismic models do not (An et al., 2015b; Shen et al., 2020; Fig 5). If the seismic models are
correct, then the high GHF estimates of the magnetic model reflect thinning of the magnetic crust, but GHF has
subsequently reduced in the ~90 My since the dominant phase of WARS crustal extension in the Cretaceous
(~105-95 Ma, Siddoway, 2008; as illustrated in Fig. 9b). If the magnetic model is correct, then GHF remains
elevated in response to the younger, 43-11 Ma Cenozoic phase of crustal extension (Granot and Dyment, 2018;
e.g. Fig. 9a). Subglacial hydrological modelling (Schroeder et al., 2014) supports the high GHF estimates in the
Thwaites Glacier region of the WARS. However, high-fidelity borehole estimates, and better constraints on the
nature of the geology, lithospheric architecture, and tectonic history of the WARS are required if we are to resolve
the different estimates and use other locations as analogues for verifying the modelled GHF distribution.

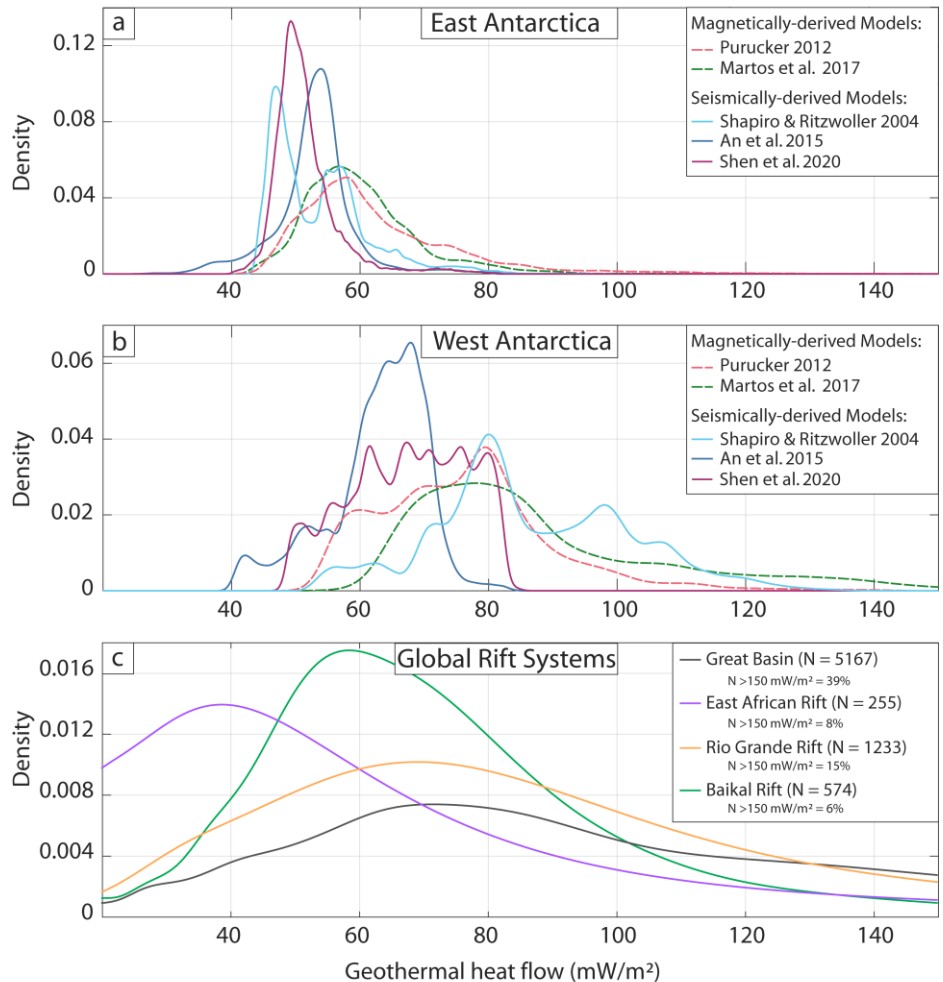


**Fig. 17. Probability density functions of the geophysically derived continental GHF datasets (Fig. 5 a – e) for a) West Antarctica and b) East Antarctica. Values extracted at 10 km spacing. The histograms were calculated with a bin size of 1 and fitted with a non-parametric distribution in the positive domain. c) GHF estimates from measured temperature gradients for global rift systems for comparison with West Antarctica (data from Lucazeau, 2019). Note the number of proportion of the data points (N) greater than 150 mW m$^{-2}$.**

To evaluate the accuracies of the different models, uncertainty estimates are required. Uncertainties of <10 mW m$^{-2}$ for the majority of Antarctica were presented for the Curie Depth GHF model of Martos et al. (2017). However, not only are the modelled values greatly different from those derived by seismic modelling (An et al., 2015b), the calculated Curie depth is deeper than the seismically- or gravitationally-derived Moho depth for large areas of the continent (Fig. 10). Even though this can occur where metallic phases are present in cratonic mantle (Ferré et al., 2013; Section 4.1.), this cannot explain the full distribution, nor are these occurrences likely to be this extensive. Without being critical of the model itself, it is reasonable to dispute the accuracy of the calculated uncertainties, and suggest that although their calculation from the geophysical data may be logical, there may be a geological contribution to uncertainty (e.g. lithological variation in the lithosphere) that is not being considered. As GHF models are utilised by researchers in different fields to those publishing the models, they cannot be independently evaluated by the user, and so accuracy in published uncertainty values is arguably more important than the accuracy of the model itself. We recommend that future research (including geophysical, geological, glaciological, and borehole and probe-derived estimates) is careful in its presentation of uncertainty.

The largest limitations to existing geophysical-derived GHF models are uncertainties in the structure,
composition, heat production, and thermophysical properties of the unexposed crust, lithosphere, and underlying
mantle. All current continental models assume the lithosphere to be laterally homogenous in its composition and
thermophysical properties, and although seismic GHF models (e.g. An et al., 2015b) incorporate variable mantle
temperatures, its composition is assumed to be homogenous. Geophysical GHF models assume that lithospheric
heat production is focussed in the upper crust, and is orders of magnitude greater than the deeper heat production
of the middle and lower crust and the mantle. These models assume that lithospheric heat production either
exponentially decreases with depth (e.g. the Curie depth models of Fox Maule et al., 2005; Martos et al., 2017;
and Purucker, 2012) or is concentrated within a laterally homogenous layer of variable depth and constant heat
production (e.g. the seismic model of An et al., 2015a, and the thermal-isostatic model for Australia of Hasterok
and Gard, 2016). However, although the lower crust is enriched in mafic rocks (iron-rich rocks of high
crystallisation temperature, e.g. basalt) of low heat production, deep boreholes and crustal sections have shown
that whilst there is a correlation between heat production and lithology in the upper crust, there is no such
correlation with depth or metamorphic grade (Section 4.6.3.). Similarly, the assumption of laterally homogenous
heat production has been shown to be unreasonable for estimation of Antarctica's GHF, which (like all continents)
has a laterally variable geology and associated concentration of HPEs (Burton-Johnson et al., 2017; Carson et al.,
2014). The exponential decrease model of crustal heat production should thus be rejected, and attempts should be
made to derive the depth and structure of crustal heat production.
The most promising approach to address the challenge of uncertainty in the contribution to GHF from the
unexposed crust and deeper lithosphere is the derivation of a three-dimensional lithospheric structure model for
Antarctica. This approach uses geophysical modelling integrating seismic, magnetic, and thermal-isostatic
evidence, and integrating into the modelling the heat production, conductivity, and petrophysical properties of
exposed lithologies and deeper crustal xenoliths or crustal sections. A similar model was developed for Norway
(Ebbing et al., 2006; Olesen et al., 2007), and an Antarctic model would build upon recent 2D and 3D
geophysically-derived models (Leat et al., 2018; Pappa et al., 2019b, 2019a). This requires an expanded database
of the geochemistry of Antarctica's rock outcrops (particularly the HPEs). Beneath the Antarctic ice sheet, where
the surface geology is unknown, the lithologies and probable heat production is best constrained by determining
the probable heat production of each drainage basin based on its detrital clasts (e.g. Goodge, 2018).
The assumption of a homogenous mantle composition beneath East Antarctica is challenged by discrepancies
between the Moho depth models derived by gravity and isostatic modelling (Pappa et al., 2019b, 2019a), as this
indicates variable lithospheric mantle densities, or deeper mantle effects on topography. A review of the available
mantle xenoliths and mantle-derived basalt chemistry may be able to constrain the composition of the mantle
beneath Antarctica, and thermal-isostatic modelling may be able to identify these regions of anomalous mantle
anomalies (as in the Australian study of Hasterok and Gard, 2016). If the seismic data for Antarctica is sufficient
to determine crustal density, such a thermal isostatic model would provide an additional independent method to
determine the depth of the upper crustal heat producing layer (Hasterok and Chapman, 2011) and evaluate the
other GHF models.
Finally, it is important to compare Antarctica with its conjugate margins (e.g. Pollett et al., 2019), where GHF and
crustal structure are better constrained. This provides constraints on the GHF along the margins of East Antarctica,
as well as informing on the geology beneath the ice sheet.
Beyond individual geological and geophysical approaches, a further challenge is how best to integrate different
models. Robust methods must be developed to incorporate datasets with different resolutions and uncertainty,
including techniques already used by the broader data analysis community. For example, Rezvanbehbahani et al.
(2017) applied a multi-variate regression analysis to estimate heat flow in Greenland from sparse and variable
geological and geophysical models and data.
7.3. **Glaciological GHF estimates**
Englacial temperatures are more sensitive to GHF in areas of the interior of Antarctica where basal sliding is
negligible (Section 2.1). Out of all the methods discussed to derive GHF in the Antarctic interior, the most
promising method is to derive GHF from englacial temperatures obtained from microwave emission (Section 5.4.)
at a longer wavelength (0.5 GHz) that the currently available (~1.4 GHz). The increase in wavelength will reduce
the uncertainty in englacial temperatures below 1000-1500 m (Jezek et al., 2014). By improving the estimations
of englacial temperature near the bed, this will reduce the role of ice flow modelling required to extrapolate
temperature from the partial-depth data. Potentially, if near-the-bed englacial temperatures are known with
sufficient precision, GHF could be derived as from borehole thermometry (Section 3.2). However, the longer-
wavelength method requires the acquisition of currently unavailable satellite-derived data. The method is only
applicable in areas of thick, very slow-flowing ice, and within this area only two ice boreholes exist for validation.
Further validation of the technique to determine the origin of the differences between the temperature model
derived from emissivity data, and glaciological thermal modelling (Macelloni et al., 2019), and other spatially
variable processes affecting microwave emissivity must also be considered (e.g. wind speed, accumulation rate,
surfaces roughness, and density heterogeneities in the firn layer; Passalacqua et al., 2018).
Existing glaciological data, like subglacial water distribution or dated englacial layers, has been successfully used
in estimating heat flow in regions of thick, slow flowing ice near ice divides, where advection and shear heating
are minimised. To extend these regional studies to continental scale, both data and models have to be improved.
A significant challenge for radar-derived subglacial water distribution is our ability to discriminate between water
at the bed versus contrasts in the geometric properties of ice sheet and bed (Schroeder et al., 2014). However the
improvement in radar techniques and the combination with seismic surveys and direct access observations, is our
best chance to improve our observations of subglacial hydrology (Ashmore and Bingham, 2014).
The inventory of subglacial lakes (Wright and Siegert, 2012) is a better constrained and expanding dataset.
Subglacial lakes can be detected also using satellite surface altimetry (Fricker et al., 2007), providing a way to
expand the coverage and to confirm dubious cases. However, as noted in Section 5.1., topography must be
considered when using evidence for subglacial lakes as they can only develop in topographic depressions, and the
absence of basal water does not imply the bed is frozen if water can drain away.
Subglacial melting can also be detected in englacial stratigraphy (Section 5.3) but the required radar product
(internal radar reflective horizons) is not often available. "AntArchitecture" is a SCAR (Scientific Committee on
Antarctic Research) Action Group bringing together key datasets on Antarctic internal layering from the principal
institutions and scientists who have been responsible for acquiring, processing and storing them over the last four
decades (AntArchitecture Action Group, 2017). As the coverage of Antarctic internal layers becomes widely
available, its application to infer GHF will increase in popularity.
Finally, and for any of the glaciological methods described above, the glaciological models used to infer GHF
have to be improved. The current thermal models used to infer GHF can be classified in two larger groups: 1) 1D
time-dependent high-complexity models, and 2) 2D/3D steady-state low-complexity models. The first category is
generally used near ice domes or ridges, with low horizontal flow, and where horizontal heat advection can be
neglected (e.g., Passalacqua, 2017). The latter are used across the whole continent (e.g. Van Liefferinge et al.,
2018), but ignore the changes in temperature between glacial and interglacial periods despite their strong effect
on englacial temperatures (Ritz, 1989). The challenge is to develop thermal models with the required level of
complexity at a continental scale, accommodating the main physical processes. This remains a technical challenge,
and thermodynamic models remain dependant on GHF estimates.

## 8. Conclusions

We present state-of-the-art data and models to estimate geothermal heat flow in Antarctica and highlight the need
for a detailed continental map. We also discuss current challenges and future directions.
With multiple methodologies and models for Antarctic GHF currently published, the most promising future
direction for local estimates is borehole-derived estimation of GHF beneath the Antarctic ice sheet from RAID
bedrock drilling and englacial temperatures from ice boreholes. Ideally, the latter approach will be validated by
the former to support expansion of the dataset from shallow boreholes (potentially only 600 m deep, or 20 % of
the total ice sheet thickness).
The ice sheet is most sensitive to variation in GHF within the interior of Antarctica, where heat production from
sliding at the base of the ice sheet is negligible. However, it is in this region that GHF is hardest to constrain by
geophysical estimates because is of the scarcity of local GHF estimates from down-hole measured temperature
gradients, geological data, and insight from conjugate margins. It is thus in the interior of Antarctica where
glaciological approaches are the most applicable. Out of the methods presented, the determination of englacial
temperatures from long-wavelength microwave emissivity is the most promising, but this data is not currently
available and requires further validation.
We highlight the potential of regional estimates of GHF from subglacial meltwater inventories. Aside from the
ever expanding inventory of subglacial lakes we encourage initiatives like "AntArchitecture" that will make radar
products widely available. Also, we discuss future requirements of thermal models (either 1D or those lacking
glacial-interglacial variability) to expand the methods beyond domes in the interior of Antarctica.
Geophysical methods remain the most attractive approach to estimate GHF because they are independent of ice
flow. However, they vary greatly in their estimated magnitude and distribution of GHF. The greatest uncertainty
in all the geophysical models is uncertainty in the composition and structure of the lithosphere and mantle. We
recommend ceasing to use the exponential decrease model of crustal heat production. Instead, we suggest using
geological and geophysical approaches to model the thickness, structure and composition of the crust. We also

recommend the application of a thermal-isostatic approach to provide an independent estimate, and highlight regions of anomalous isostatic elevation and probable mantle heterogeneities. The effects of topography must also be considered in all GHF models.

Finally, the greatest challenge for Antarctic GHF estimation is the necessity for multidisciplinary science and how best to integrate the different methods. Hopefully, this paper provides a first step in communicating the approaches and limitations of the different fields across the GHF community. We sincerely recommend the continuation and enhancement of the international collaborations within SCAR, building on the work of the GHF sub-group of the SERCE research programme (Solid Earth Response and influence on Cryospheric Evolution), and encourage and appreciate SCAR's continuing support in this field of research.

**Data availability**

The database of GHF point estimates (Fig. 15) is available in the Supplementary Material and from https://github.com/RicardaDziadek/Antarctic-GHF-DB. The GHF mean and standard deviation maps of the geophysical models of continental GHF (Fig. 5f and 5g) are available in the Supplementary Material.

**Author contributions**

ABJ and RD conceived the project. ABJ was the lead author for all sections and the specialist in geology and geochemistry. RD was a co-author for all sections, compiler of the supplemental GHF database, and the specialist in geophysics. CM was a co-author for all sections and the specialist in glaciology.

**Competing interests**

The authors declare that they have no conflicts of interest.

**Acknowledgements**

The authors thank Brice Van Liefferinge, an anonymous reviewer, and John Goodge for their helpful and thorough reviews and comments, which have all improved this final manuscript, Alexander Robinson for his work and comments as our handling editor, and the Copernicus team for all their assistance. This research is a contribution to the SCAR SERCE scientific research programme, and we thank the discussions and support of this group from the TACtical 2018 (Hobart, Australia), POLAR 2018 (Davos, Switzerland), ISAES 2019 (Incheon, Korea), and SCAR 2020 Open Science Conference (online) meetings. We particularly thank Jacqueline Halpin (IMAS, Hobart) for her comments on the manuscript and her work promoting and developing the Antarctic GHF community.

**Financial support**

A. Burton-Johnson and C. Martin were funded by the Natural Environment Research Council as part of the British Antarctic Survey Polar Science for Planet Earth programme. R. Dziadek was supported by the Deutsche Forschungsgemeinschaft (DFG) in the framework of the Priority Program 1158 "Antarctic research with comparative investigations in Arctic ice areas" by grant GO 724/14-1. Additional funds were contributed by the AWI Research Program PACES-II Workpackage 3.2.

**Review statement**
This paper was reviewed by Brice Van Liefferinge and an anonymous reviewer. Alexander Robinson was the
handling editor. John Goodge provided an additional short comment in the interactive discussion.

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
