# Peer review of "Geothermal heat flow in Antarctica: current and future"

_The Cryosphere, 2020_

## Short Comment (SC1) · 25 Mar 2020

Nice paper and worthwhile compilation of ideas as well as a look forward. I have a few suggestions, mainly to help improve organization of topics.

I wonder about the overall organization of section 4.6, which is about how we make GHF estimates in heterogeneous crust. The opening section 4.6.1 goes into determining heat production from rock samples obtained from exposure, but does not discuss interpretations of GHF offered in these papers. On the other hand, GHF is discussed in sections 4.6.3 and 4.6.4, building on other ways to get at heat production. Seems perhaps better to comment on the implications for GHF from the heat production studies and how this reflects heterogeneities?

[Figure]

Further, I understand the distinction between rock outcrop and sampling rocks from moraines, but I wonder if it would make more sense to move up the discussion of glacial moraine materials from the CTM either into section 4.6.1 or perhaps changing that clast section to follow the other as new section 4.6.2? They both relate to determining heat production in rocks.

Also, I suggest changing the title of section 4.6.4 from 'Detrital material' to 'Glacially-derived rock clasts' or something along those lines. For better or worse, 'detrital material' to many people will conjure up detrital minerals from sedimentary deposits or sedimentary rocks, or even sediment itself. In this case, it's an important distinction because we sampled large rock clasts that can be treated analytically just like any rock samples taken from exposure.

John Goodge

---

## Referee Comment (RC1) · Brice Van Liefferinge (Referee) · 13 Apr 2020

Dear Authors,

I enjoyed reading your paper on the comparison of the different GHF estimate methods. This paper provides a great overview of the work done on GHF reconstructions and provides key future directions. All known methods are described and are well supported by explicit examples and references in the manuscript. I really liked section 4 on GHF derived estimates. I think that all the key references are included. I added a few and strongly suggest to add and describe the work of Rezvanbehbahani et al. (2017) on machine learning techniques as done in Greenland. The introduction and conclusion support well the manuscript as do the figures (see specific comments).

The language used is appropriate. However, I would suggest the following general changes: a) In the title you use "flow" but I would suggest to use "flux", as well as for the whole manuscript (see specific comments). b) The manuscript is qualitative in a few paragraphs. More quantitative descriptions could be provided such as maximum and minimum GHF of the different data sets, discuss the representativeness of point GHF values, . . . where possible. c) The limitations of the different methods to estimate GHF (ice borehole measurements, model estimates, . . .) are not always discussed. A sentence could be provided for each. E.g. ice borehole measurements provide a minimum GHF value when the base is at the pressure melting point. d) Figure 6 needs to be discussed in more detail in the text. A description of the different data sets used is lacking (see specific comments). e) Section 4.5 is quite long compared to the other subsections, and describes in a lot of details a technique that is not widely in Antarctica because of the lack of measurements. It seems therefore that including specific equations is superfluous, and perhaps the paragraph on its application in Antarctica could then be extended. I attach a detailed review of the paper for the specific line-by-line comments, see attached PDF. All the best, Brice Van Liefferinge

Please also note the supplement to this comment:
https://www.the-cryosphere-discuss.net/tc-2020-59/tc-2020-59-RC1-supplement.pdf

**Supplement:**

[Figure]

**Geothermal heat flow**
[Figure]
 in Antarctica: current and future directions

Alex Burton-Johnson[1], Ricarda Dziadek[2], and Carlos Martin[1]

[1]British Antarctic Survey, High Cross, Madingley Road, Cambridge, CB3 0ET, UK

[2]Alfred Wegener Institute - Helmholtz Centre for Polar and Marine Research, Am Alten Hafen, Bremerhaven, Germany

*Correspondence to:* Alex Burton-Johnson (alerto@bas.ac.uk)

**1. Abstract**

Antarctic geothermal heat flow (GHF) affects the temperature of the ice sheet, determining its ability to slide and internally deform, as well as the behaviour of the continental crust. However, GHF remains poorly constrained, with few and sparse local, borehole-derived estimates, and large discrepancies in the magnitude and distribution of existing continent-scale estimates from geophysical models. We review the methods to  GHF, compile borehole and probe-derived estimates from measured temperature profiles, and recommend the following future directions: 1) Obtain more borehole-derived estimates from the subglacial bedrock and englacial temperature profiles. 2) Estimate GHF beneath the interior of the East Antarctic Ice Sheet (the region most sensitive to GHF variation) via long-wavelength microwave emissivity. 3) Estimate GHF from inverse glaciological modelling, constrained by evidence for basal melting. 4) Revise geophysically-derived GHF estimates using a combination of Curie depth, seismic, and thermal isostasy models. 5) Integrate in these geophysical approaches a more accurate model of the structure and distribution of heat production elements within the crust, and considering heterogeneities in the underlying mantle. And 6) continue international interdisciplinary communication and data access.

**Summary of Comments on tc-2020-59_BVL.pdf**

**Page: 1**

Number: 1        Author: brice        Subject: Sticky Note        Date: 13/04/2020 11:20:51
This might seem to be a picky comment but it is important to be precise. I would suggest to use "flux" than "flow": heat flow should be reserved for the movement of material while heat flux is a transport of a quantity of energy over time.  As in this paper you focus more on the GHF beneath the Ice Sheet (bedrock surface), I would use "flux". Otherwise can you explain the use of "flow" ?

Number: 2        Author: brice        Subject: Inserted Text        Date: 13/04/2020 11:20:59
"estimate" (extract sounds like by effort or force)

**1. Introduction**

The Antarctic ice sheet is the world's largest potential driver of sea level rise, and accurately modelling its dynamics relies, amongst others, on constraining conditions at the ice-bedrock interface. Measuring these basal conditions is inherently challenging and, of all the parameters affecting ice sheet dynamics, subglacial geothermal heat flow (GHF) is the least constrained (Larour et al., 2012; Llubes et al., 2006). Despite this uncertainty, GHF affects (1) ice temperature and, as a consequence, ice mechanical properties (rheology), (2) basal melting and sliding, and (3) the development of unconsolidated water-saturated sediments; all of which can promote ice flow (Greve and Hutter, 1995; Larour et al., 2012; Siegert, 2000; Winsborrow et al., 2010). Beyond ice dynamics, our knowledge of GHF allows us to model past basal melt rates in our exploration for old ice core climate records,
 constrain models of glacial isostatic adjustment (GIA), and inform on the geological and tectonic development of Antarctica.

In recognition of the ambiguity and importance of Antarctic GHF, an increasing number of studies in geology, geophysics, and glaciology have sought to constrain this parameter, with a developing dedicated multinational interdisciplinary community (Burton-Johnson et al., 2019; Halpin and Reading, 2018). However, with an expanding research base and a requirement for multidisciplinary science, the necessity for a multidisciplinary review of current approaches and future directions was highlighted by the GHF sub-group of SERCE (Solid Earth Response and influence on Cryospheric Evolution) and the Scientific Committee on Antarctic Research (SCAR) (Burton-Johnson et al., 2019).

**1.1. What is geothermal heat flow (GHF)?**
[Figure]

GHF describes the transport of heat energy from the interior of the Earth to the surface (Gutenberg, 1959; Pollack et al., 1993). This heat originates from two primary sources: 1) The primordial heat remaining from the formation of the Earth, when the kinetic energy of celestial collisions was transformed into heat energy; and 2) the radioactive decay of heat-producing elements (HPEs) and their isotopes; 98% of which is derived from Uranium, Thorium, and Potassium (Beardsmore and Cull, 2001; Lowrie, 2007). The HPEs are incompatible with the mineral structures of the mantle, so are concentrated into the crust (Boden, 2016; McDonough and Sun, 1995). Other sources of possible contributions to GHF are: 1) geoneutrino emission from the mantle (Huang et al., 2013; Korenaga, 2011), and 2) gravitational pressure (Elbeze, 2013; Morgan et al., 2016).

The estimated average heat flow of continental crust is 67.1 mW m$^{-2}$, whilst for oceanic crust it is 78.8 mW m$^{-2}$ (Lucazeau, 2019; although estimates vary according to sampling strategy and the number of observations). The difference between continental and oceanic heat flow reflects the lower thickness of oceanic crust, with hot mantle rocks at comparatively shallow depths. Continental GHF varies significantly, primarily in response to variations in crustal heat production, age, composition, tectonic history, and thickness of crust and mantle (Mareschal and Jaupart, 2013). This results from the geological complexity of composite continental crust compared with oceanic crust. GHF is generally lower in stable crust away from convergent and divergent continental margins and rift basins, and higher in these magmatically active provinces (Lucazeau, 2019; Pollack et al., 1993). On a broad regional scale, continental GHF correlates negatively with age, allowing first order empirical estimation of Antarctic GHF based on its range of crustal ages (Fig. 1; Llubes et al., 2006; Sclater et al., 1980). However, Antarctic crustal heat production estimates show high variability across sampled age ranges (Gard et al., 2019),

**Page: 2**

| | | | |
|---|---|---|---|
| Number: 1 | Author: brice | Subject: Inserted Text | Date: 13/04/2020 11:21:24 |

present and

| | | | |
|---|---|---|---|
| Number: 2 | Author: brice | Subject: Sticky Note | Date: 13/04/2020 11:21:30 |

See comment on the Title

[revised manuscript text omitted]

I would suggest to add somewhere that ice sheet temperature is also influenced by the ice thickness: as ice acts as an insulator, the greater the ice thickness, the warmer the ice at the base. This is counterbalanced by cold temperature advecting from the surface, itself influenced the accumulation rate.
* * *
**Number: 2**     Author: brice     Subject: Inserted Text     Date: 06/04/2020 15:43:05

whole
* * *
**Number: 3**     Author: brice     Subject: Inserted Text     Date: 06/04/2020 15:43:59

basal
* * *
**Number: 4**     Author: brice     Subject: Cross-Out     Date: 06/04/2020 15:59:01
* * *
**Number: 5**     Author: brice     Subject: Inserted Text     Date: 06/04/2020 15:58:43

"increase" ==> from -13°C to 7°C
* * *
**Number: 6**     Author: brice     Subject: Inserted Text     Date: 13/04/2020 11:23:23

and expands the surface area of the bed at the pressure point from 16% to more than 50%.
* * *
**Number: 7**     Author: brice     Subject: Cross-Out     Date: 06/04/2020 16:01:57
* * *
**Number: 8**     Author: brice     Subject: Inserted Text     Date: 13/04/2020 11:23:44

As you mention surface temperature in this paragraph, I suggest to add a sentence on surface accumulation which can have a strong influence on the basal conditions even in the interior of the ice sheet, and counteract the effect of the GHF (see Fig. 2 Van Liefferinge and Pattyn 2013)

[Figure]

In addition to enhancing basal melting and reducing basal friction, increased GHF enhances ice flow by increasing the englacial temperature and thus reducing the ice stiffness (Larour et al., 2012). Because the heat produced by basal friction and viscous deformation are orders of magnitude greater than from GHF in fast-flowing ice streams, this effect is only significant in upstream, slow-flowing areas (Larour et al., 2012). In these regions of thick, slow- flowing ice, even local high heat flow anomalies of insufficient heat for basal melting can result in the development of accelerated, channelised flow for hundreds of kilometres upstream and downstream of the GHF

anomaly (Pittard et al., 2016a). Regions along ice divides and adjacent to ice streams are particularly sensitive to enhanced GHF (Pittard et al., 2016b).

Whilst the points above highlight the necessity of estimating Antarctic GHF, it is very important that the accuracy of these estimates can be verified. The impact of inaccurate GHF constraints on models of ice sheet dynamics have been shown by comparing GHF estimates for Greenland. Ice sheet modelling controlled by spatially variable

GHF forcing reproduces the observed state to only a limited degree, and fails to reproduce either the topography or the low basal temperatures measured in southern Greenland (Rogozhina et al., 2012). Instead, an unrealistic spatially uniform GHF forcing produces a considerably better fit. If the much larger Antarctic ice sheet is to be accurately modelled, the accuracy of the GHF estimates used must be well constrained by multiple independent methodologies, sensitivity tests, and comparison of different models.

Recently, there has been increasing interest in the exploration of suitable locations for coring Antarctica's oldest continuous ice record. This problem requires accurate knowledge of GHF, as basal melt rates limit the maximum possible age of recoverable ice (Fifferinge et al., 2018). Additionally, due to environmental concerns around possible drilling fluid contamination, frozen bed conditions are a prerequisite for deep coring operations.

2.2. **Glacial Isostatic Adjustment (GIA)**

The temperature of the lithosphere and upper mantle are important parameters for modelling the isostatic response to changes in the volume of the overlying ice sheet (i.e. glacial isostatic adjustment, GIA). This is because the (visco-)elastic properties of the lithosphere and mantle directly relate to its thermal properties (Chen et al., 2018;

Kuchar and Milne, 2015). GIA is a critical component of the long-term evolution of ice sheets and could potentially stabilise retreating ice streams in submarine settings (Barletta et al., 2018; Kingslake et al., 2018). Of particular importance here is that the temperature-dependant viscosity that controls GIA can be modelled using surface heat flow estimates (van der Wal et al., 2013, 2015).

**Page: 5**

Number: 1     Author: brice     Subject: Inserted Text     Date: 13/04/2020 11:24:25

You should also add the work of Rezvanbehbahani et al. (2017) : Rezvanbehbahani et al. (2017) use for the first time machine learning techniques to derive GHF from relevant geologic features (gravity measurements, magnetic anomaly) and  GHF measurements (derived from crustal thickness, rock composition and active thermal feature).

S. Rezvanbehbahani, L. A. Stearns, A. Kadivar, J. D. Walker, and C. J. van der Veen. Predicting the geothermal heat flux in Greenland: a machine learning approach. Geophysical Research Letters, 2017. ISSN 1944-8007. doi: 10.1002/2017GL075661.

Number: 2     Author: brice     Subject: Inserted Text     Date: 06/04/2020 16:33:34

Cit: Fischer 2013, Climate of the past paper. https://www.clim-past.net/9/2489/2013/

Number: 3     Author: brice     Subject: Inserted Text     Date: 06/04/2020 16:12:44

Van Liefferinge and not Liefferinge

## 2.3. Geology and tectonics

[Figure]

**Fig. 2. Basic illustration of a subduction zone at a convergent margin between oceanic and continental lithosphere to clarify the geological concepts and terms used in this paper.**

**2.3.1. Mantle dynamics**

Heat flow variation and its isostatic effects (i.e. the buoyancy control on crustal elevation, resulting from the different densities of the dense mantle and less dense overlying crust) provide evidence for mantle dynamics beneath a continent. For example, high heat flow anomalies have been proposed as evidence for sub-lithospheric heating by present and past mantle plumes (regional hot spots of warm mantle upwelling beneath the lithosphere; e.g. Courtney and White, 1986; Martos et al., 2018) the absence of enhanced heat flow where mantle ascent is proposed has been used to argue against such processes (e.g. Stein and Stein, 2003). Also, because of the relationship between surface heat flow and isostatic elevation, heat flow studies can reveal thermal or compositional variation of the sub-continental mantle, as a reduction in its density can increase the isostatic elevation of the surface topography (Hasterok and Gard, 2016).

**2.3.2. Development of the lithosphere**

The thermal properties of the lithosphere control its response to tectonic deformation (e.g. Sandiford and Hand, 1998), such as the development of crustal shear zones and earthquakes. The lithosphere's thermal properties also affect the relative density of lithosphere and underlying mantle, and (as a result of this buoyancy effect) the isostatic surface elevation. This in turn influences the heights of Antarctica's mountain ranges and the depths of its sedimentary basins (McKenzie et al., 2005). For these reasons, understanding the continent's GHF will inform on the development of many of Antarctica's largest tectonic features. For example, the lithospheric extension of the West Antarctic Rift System, the prominent elevation of the Transantarctic Mountains, the deep topographic depression of the Wilkes subglacial basin, and the extensive Palmer Land Shear Zone of the Antarctic Peninsula.

**3. GHF estimates from measured temperature gradients**

Having highlighted the importance of constraining Antarctica's GHF, the following sections discuss current approaches to its estimation.
* * *
**Number: 1**     Author: brice     Subject: Sticky Note     Date: 10/04/2020 11:53:16

Curie Point Depth (CPD as in section 4.1 L311)
* * *
**Number: 2**     Author: brice     Subject: Sticky Note     Date: 10/04/2020 11:06:55

I suggest to develop in one or two sentences the implications of the Antarctic Ice Sheet like in section 2.3.2
* * *
**Number: 3**     Author: brice     Subject: Inserted Text     Date: 13/04/2020 11:25:39

It is a simple suggestion but why not provide a table presenting all the methods used to estimate the GHF with the advantages and disadvantages, to have an overview of all the methods together. This sentence could be extended as well to give an overview of the section's content.

[revised manuscript text omitted]

* * *
**Number: 1**     Author: brice     Subject: Inserted Text     Date: 13/04/2020 11:26:05

In the xlsx supplementary material, I guess that in the row "method", borehole means "Boreholes into bedrock" ? If yes can you provide the exhaustive (or estimation) of the number of boreholes into bedrock in Antarctica and cite the SOM
* * *
**Number: 2**     Author: brice     Subject: Underline     Date: 13/04/2020 11:26:38

A key point that is not explained here, is that, when the base of the ice sheet is at the pressure melting point (presence of water), the GHF estimate is a minimum GHF estimate, which means that the GHF can be higher! See also section 5, 5.1
* * *
**Number: 3**     Author: brice     Subject: Inserted Text     Date: 10/04/2020 11:26:41

add citation

[revised manuscript text omitted]

Number: 1     Author: brice     Subject: Sticky Note     Date: 13/04/2020 11:27:21

A: Fox Maule et al., did you use the 2005 version of the data set or the updated one from Purucker ? Based on the figure, you are using the Purucker et al update so please cite:

M. Purucker. Geothermal heat flux data set based on low resolution observations collected
by the champ satellite between 2000 and 2010, and produced from the mf-6 model following the technique described in fox maule et al.(2005). See http://websrv. cs. umt. edu/isis/index. php, 2013.

[Figure]

[Figure]

**Fig. 6. Coverage of sub-continental scale regional estimates of GHF.**
[Figure]

4.1. **Magnetic ** [2]

As for the penetrative methods of GHF estimation described above (Section 3), geophysical methods also derive

GHF from a temperature gradient. In this case, magnetic survey data is used to determine the depth at which the maximum temperature of ferromagnetic magnetisation is exceeded (the Curie temperature; Haggerty, 1978). This

Curie temperature is different for different minerals, but is assumed in these studies to the Curie temperature of magnetite (580 °C) as this mineral is most commonly the dominant contributor to crustal magnetisation (Bansal et al., 2011; Fox Maule et al., 2005; Langel and Hinze, 1998).

Above the Curie temperature, rocks lose their ability to maintain ferromagnetic magnetisation (e.g. Haggerty,

1978). The depth of this isotherm in the crust (the Curie Point Depth, CPD; Fig. 7 and Fig. 2) is thus assumed to be the depth to the bottom of the magnetic source (DBMS) determined from magnetic survey data. The DBMS

maps a transition zone, rather than an exact depth (Haggerty, 1978), and can provide information on crustal temperatures at depths not accessible by other means (Andrés et al., 2018; Okubo et al., 1985). Regions found to have a shallower DBMS (and thus an assumed shallower CPD) are expected to have higher average temperature gradients, and, therefore, higher GHF (e.g. Aboud et al., 2011; Andrés et al., 2018; Arnaiz-Rodríguez and

Orihuela, 2013; Bansal et al., 2013, 2011; Bhattacharyya and Leu, 1975; Guimarães et al., 2013; Li et al., 2017;

Obande et al., 2014; Okubo et al., 1985; Ross et al., 2006; Salem et al., 2014; Tanaka et al., 1999; Trifonova et al., 2009).

Number: 1     Author: brice     Subject: Inserted Text     Date: 13/04/2020 11:27:49

I would suggest to be very explicit about what each data set is in that figure. e.g. Passalacqua et al., 2017: "radar reflectivity and inverse modelling". Provide a table with the links to the data sets ?
(see general comments)

Number: 2     Author: brice     Subject: Inserted Text     Date: 10/04/2020 15:51:30

derived estimates (to be consistent with the other sub-section title )

[Figure]

**Fig. 7. Approximation of the geothermal gradient from the Curie point depth (CPD).  1 assumed to mark the base of the magnetic crust (DBMS).**

[revised manuscript text omitted]

. B)

| | Number: 2 | Author: brice | Subject: Cross-Out | Date: 10/04/2020 11:59:10 |

| | Number: 3 | Author: brice | Subject: Inserted Text | Date: 10/04/2020 11:59:26 |

highlighted magnetic assemblages of Fe-Ni-Co-Cu metal alloys up to 620 °C to 1084 °C (Haggerty, 1978). Without further constraints and validations, these assumptions remain the best approach, especially in sparsely sampled regions like Antarctica, but introduce uncertainties of several kilometres in Curie depths and consequent uncertainties in

GHF estimates (Bansal et al., 2011; Ravat et al., 2007). Similarly, in areas of thin crust, non-magnetic mantle rocks can be shallower than the Curie depth. In these regions, the calculated Curie isotherm will appear shallower due to a lack of magnetic minerals in the mantle rocks (Fig. 9.; Frost and Shive, 1986; Wasilewski and Mayhew,

1992). This can be investigated through comparison of the Antarctic Curie depth estimates with the seismically- or gravitationally-derived depth of the crust-mantle boundary (the Moho depth; Fig. 10 and Fig. 2).

[Figure]

**Fig. 9. Two scenarios illustrating the ambiguity in estimating Curie isotherm depth and GHF. a) Estimates from a**

**region with a shallow Curie isotherm over an area of thin crust. b) Similar but incorrectly interpreted estimates from**

**a region of shallow non-magnetic mantle rocks. In scenario (b), the DBMS is shallower despite there being no deviation**

**in the Curie isotherm depth. DBMS = depth to the Bottom of the Magnetic Source, assumed to represent the Curie**

**depth in the GHF estimates discussed).**

[Figure]

**Fig. 10. Comparison of Curie depth (Martos et al., 2017) and depth of the crust-mantle boundary (the Moho depth)**

**derived from a) gravity modelling (Pappa et al., 2019b), and b) seismic modelling (An et al., 2015a). Negative values**

**show areas where the estimated Curie depth is deeper than the estimated Moho depth, and positive values are where**

**the Curie depth is shallower than the Moho depth.**
* * *
**Number: 1**     **Author:** brice     **Subject:** Sticky Note     **Date:** 10/04/2020 12:02:28

CPD
* * *
**Number: 2**     **Author:** brice     **Subject:** Cross-Out     **Date:** 10/04/2020 12:05:04
* * *
**Number: 3**     **Author:** brice     **Subject:** Inserted Text     **Date:** 10/04/2020 12:05:06

:
* * *
**Number: 4**     **Author:** brice     **Subject:** Inserted Text     **Date:** 10/04/2020 12:05:00

bottom of the magnetic source
* * *
**Number: 5**     **Author:** brice     **Subject:** Sticky Note     **Date:** 13/04/2020 11:29:37

As is, to me, this figure illustrates more the difference between Martos et al. (2017) and the two methods than between seismic and gravity modeling. Why not simply plot the difference between the two methods you describe?

[revised manuscript text omitted]

| Number: 1 | Author: brice | Subject: Inserted Text | Date: 13/04/2020 11:29:47 |
Thanks for pointing this difference out !

| Number: 2 | Author: brice | Subject: Inserted Text | Date: 10/04/2020 15:53:14 |
model derived estimates (e.g)

| Number: 3 | Author: brice | Subject: Inserted Text | Date: 10/04/2020 15:53:52 |
as before for the sub-title.

northern India in the seismically derived model of East Antarctica (An et al., 2015b). However, when extrapolating heat flow away from the conjugate margins into the interior of Antarctica, this approach is susceptible to the method of interpolation used and the quality and scarcity of the borehole-derived GHF estimates in the interior of

Antarctica (Section 3).
[Figure]

[Figure]

**Fig. 11. Interpolated heat flow map of Gondwana, showing the derivation of Antarctic GHF from the reconstructed**

**conjugate margins of the supercontinent. Terrestrial heat flow data shown by points. Adapted from Pollett et al. (2019).**

## 4.5. Isostatic elevation

In addition to crustal thickness and density, the thermal state of the lithosphere also contributes to its isostasy and observed surface elevation. The effect of thermal isostasy on the bathymetry of oceanic crust is well recognised:

as oceanic crust migrates from the spreading ridge it cools, thickens, contracts, and subsides (Stein and Stein,

1992). However, the effect of thermal isostasy on continents is masked by compositional contributions to isostatic elevation (i.e. lateral variations in crustal thickness and density, Fig. 12a; Hasterok and Chapman, 2007b, 2007a).

**Fig. 12. Relationship of the median observed (a) and adjusted (b) elevation and median compiled heat flow values of 36**

**geological provinces on the land and continental shelves of North America, ranging from 30 - 2082 x 10$^3$ km$^2$. Compiled**

Number: 1     Author: brice     Subject: Inserted Text     Date: 10/04/2020 15:55:09
Can you provide some numbers on the maximum and minimum GHF ?

Number: 2     Author: brice     Subject: Sticky Note     Date: 13/04/2020 11:20:21
Section 4.5 is quite long compared to the other subsections, and describes in a lot of detail a technique that is not widely used in Antarctica because of the lack of measurements. It seems therefore that including specific equations is superfluous, and perhaps the paragraph on its application in Antarctica could then be extended.

[revised manuscript text omitted]

**Number: 1**   Author: brice   Subject: Underline   Date: 10/04/2020 16:14:16
You use whilst several times, you could remove some of them.

**Number: 2**   Author: brice   Subject: Underline   Date: 10/04/2020 16:15:18

**Number: 3**   Author: brice   Subject: Inserted Text   Date: 10/04/2020 16:18:25
 you can also cite: J. W. Goodge, C. M. Fanning, C. M. Fisher, and J. D. Vervoort. Proterozoic crustal evolution of central East Antarctica: Age and isotopic evidence from glacial igneous clasts,
and links with australia and laurentia. Precambrian Research, 299:151–176, 2017.

of other Precambrian cratons (>542 Ma regions of tectonically-stable continental crust; e.g. central Canada). These clasts were analysed for their HPE abundance and attributed to their likely source area (the drainage basin of their associated glaciers). A probable range of subglacial heat flow values was estimated by assuming mantle and lower crustal GHF values and a thickness for the upper crust based on other Precambrian shields. This indicates that

East Antarctic heat flow is comparable to other Precambrian cratons, and comparable to geophysical models of

East Antarctic heat flow (
efferinge and Pattyn, 2013). However, broader application of this approach is biased towards more erosion resistant rock types, whilst less competent lithologies will not be preserved after glacial transport and deposition.

**5. Glaciological inverse estimation of GHF**

Although geothermal heat flow has a geological derivation, it can also be constrained by multiple approaches through its observable effects on the overlying ice sheet. Rather than using a forward modelling approach (i.e.

determining the geological contributions and estimating their resultant heat flow), an inverse modelling approach can be applied by modelling observed glaciological properties (e.g. glacial flow and melt rates) and calculating the required heat flow. We will describe in this section different methods used in glaciology to derive GHF.

**5.1. Subglacial water**

The presence of subglacial water can be detected with a ground penetrating radar. The reflective properties of the ice-bedrock interface depend on the presence of water and, with certain caveats, radar surveys can be used to map subglacial water. In general terms, a glaciological model can then be used to ⟨2⟩timate the values of GHF that better predict where basal temperatures reach the pressure melting point and melting occurs. We will describe in this Section examples of this approach.

Carter et al. (2009) modelled the dielectric loss of radar data through the ice column around Dome C in East

Antarctica (Fig. 6) to infer the basal reflectivity and verify the presence of subglacial water. Because the temperature profile of the ice sheet is one parameter affecting dielectric loss, this approach required inference of the basal heat flow from temperature-depth modelling over the last 254 ka. The Shapiro and Ritzwoller (2004)

GHF model was used initially (see section "4.2. Seismic estimates"), but when the calculated vertical ice velocity ($m_w$) at the bed exceeded the initial melt rate ($m_T$), the GHF was modified until $m_T$ and $m_w$ were equal. This approach identified localised high GHF anomalies, but (excepting these anomalies) they calculated that 66 % of the study area was either at or near the pressure melting point (anywhere that ice is thicker than 3500 m) without invoking enhanced GHF.

Schroeder et al. (2014) modelled the spatial distribution of melt beneath the ice sheet in the Thwaites Glacier catchment (Fig. 6) by mapping the relative bed echo strength of radar data in the region and modelling the water routing required to match these observations by routing alone (without heterogeneous basal melting). These routing models were based on the radar-derived ice thickness and surface slope. The 50 selected routing models were used to model the relative melt required to reproduce the observed echo strengths of each routing model.

This relative melt model was in turn scaled to match the total melt water produced in an ice sheet model of the

Thwaites Glacier incorporating frictional melting, horizontal advection, and an assumed uniform GHF. By subtracting the frictional and advective contributions, the GHF required to produce the remaining melt could be

Number: 1     Author: brice     Subject: Cross-Out     Date: 10/04/2020 16:16:32
(Van Liefferinge and Pattyn, 2013).

Number: 2     Author: brice     Subject: Underline     Date: 13/04/2020 11:30:29
As it is difficult to know the melt rate, I would say that this is not the GHF estimate but the minimum value of the GHF estimate or the minimum GHF to reach the pressure melting point.

[Figure]

calculated. This approach predicted very high heat flow in this region (114 to >200 mW m$^{-2}$), with the highest heat flow focused around observed and inferred subglacial volcanoes.

With the aim of determining appropriate sites of low basal melting for old-ice drilling, Passalacqua et al. (2017)

also used radar evidence for basal melting and ice sheet modelling to determine GHF around Dome C (Fig. 6).

Wet and dry bed conditions were identified from radar data and ten spots were identified on bedrock topographic features marking the critical ice thickness where present basal melting becomes possible. These spots were defined as locations where the upper slopes of the bedrock topography are dry and their lee slopes are wet, with melting initiating between the two when the ice thickness passes the pressure melting point (Fig. 14). Assuming that GHF

is locally homogeneous between the two bedrock elevations, heat flow was determined by increasing its value in a 1-D heat model of the local ice thickness until basal melting occurred. These point estimates were interpolated to generate an approximate map of regional heat flow and calculate basal melt rates over the last 400 ka.

[Figure]

**Fig. 14. Illustration of how the ice thickness exceeding the pressure melting point (PMP) can be identified from radar**

**reflectivity data points, indicating the presence or absence of basal water beneath the ice sheet. Once the PMP is**

**identified, thermal modelling can estimate the required local GHF. Between the thresholds of radar reflectivities**

**representative of wet and dry basal conditions, the thermal conditions are unknown (yellow-shaded region of the**

**bedrock). Adapted from Passalacqua et al. (2017).**

Pattyn and Pattyn (2013) and Pattyn et al. (2018) used steady state and transient thermodynamic

modelling of the East Antarctic Ice Sheet to map the minimum heat flow required to raise the basal temperature above pressure melting point and generate basal melting. Whilst this was executed to identify possible sites for drilling the oldest ice in areas that are unlikely to have undergone basal melting in the last 1.5 Ma and did not produce an estimate of absolute GHF, if this approach were combined with other evidence for basal conditions above the pressure melting point (e.g. combining thermodynamic modelling with subglacial lake locations) points of minimum heat flow could be mapped.

5.2. **Subglacial lakes**

If temperatures are sufficient for basal melting, and topography depressions are suitable, subglacial lakes can develop. Subglacial lakes exhibit radio reflectivities 10-20 dB greater than the ice-bedrock boundary, allowing the current identification of ~400 lakes beneath the Antarctic ice sheet.

| | Number: 1 | Author: brice | Subject: Inserted Text | Date: 10/04/2020 16:52:15 |
| --- | --- | --- | --- | --- |

Van

| | Number: 2 | Author: brice | Subject: Inserted Text | Date: 10/04/2020 16:52:19 |
| --- | --- | --- | --- | --- |

Van

| | Number: 3 | Author: brice | Subject: Inserted Text | Date: 10/04/2020 17:01:02 |
| --- | --- | --- | --- | --- |

,

| | Number: 4 | Author: brice | Subject: Inserted Text | Date: 10/04/2020 17:07:23 |
| --- | --- | --- | --- | --- |

more than

| | Number: 5 | Author: brice | Subject: Inserted Text | Date: 10/04/2020 17:07:12 |
| --- | --- | --- | --- | --- |

cite: A. P. Wright and M. J. Siegert. A fourth inventory of Antarctic subglacial lakes. Antarctic Science, 24:659–664, 2012.

[Figure]

Whether basal temperatures are sufficient for basal melting and preservation of subglacial lakes is dependent on
ice thickness, the surface temperature and accumulation rate, heat transported through ice advection, heat
produced by internal deformation and basal sliding, and the GHF. When subglacial lakes are located near ice
divides, heat derived by horizontal advection, basal friction, and internal deformation is assumed to be minimal,
and thus the heat required to bring the base of the ice sheet above the pressure melting point is a product of ice
thickness and GHF. Thus, when subglacial lakes are located near ice divides and the accumulation rate is known
(high accumulation rates cool the ice mass), point estimates of *minimum* GHF can be calculated from one-
dimensional thermal models of the ice sheet temperature profile, but an assumption that water was derived locally
and not routed from elsewhere must also be considered as lakes can only form in topographic depressions. The
absence of a lake or basal water does not imply the bed is frozen if the water can drain away (Pattyn, 2010; Siegert
and Dowdeswell, 1996).

## 5.3. Englacial stratigraphy

Jordan et al. (2018) identified draw down of internal ice sheet layers and increased bed reflectivity from radar data
near the South Pole (Fig. 6), indicating enhanced basal melting. Melt rates were calculated using dated radar
layers, traced from the Dome C ice-core site, and a depth age model that simulates the draw-down effect of ice
from subglacial melt rate. The low ice velocity ($<1.5$ m a$^{-1}$) indicated minimal frictional contribution to basal
temperature, and a location at the top of a hydraulic catchment area indicated a low heat contribution from
subglacial water. By negating these contributions to heat flow, assuming the basal temperature is at the pressure
melting point (and thus could be derived from the ice thickness) and that temporal temperature variations match
those of the Dome C ice core, a time-dependent heat equation was applied to the ice sheet to derive the basal GHF
required to generate the enhanced melt rates. [1]

## 5.4. Microwave emissivity

Englacial temperature profiles have been derived from satellite and airborne passive detection of high frequency
L-band microwave radiation (~1.4 GHz; Macelloni et al., 2019, 2016; Passalacqua et al., 2018); data primarily
collected to investigate soil moisture and ocean salinity (Kerr et al., 2010). These wavelengths have very low
absorption in ice and low scattering by particles (e.g. grainsize and ice bubbles), providing high penetration depths
in dry ice.

Macelloni et al. (2019) derived englacial temperature profiles for the Antarctic ice sheet from 2-year averaged
vertical-polarised (V) radiation collected at the "Brewster angle" (57.1° ±2.6°; the angle of incidence at which the
radiation is perfectly transmitted through the air-snow interface with no reflection, minimising the influence of
surface or shallow sub-surface effects). The corrected intensity (brightness temperature, $T_B$) correlates with the
surface temperature of the ice, but is also affected by the ice sheet thickness (a largely inverse correlation), density
profile, and grain size (Macelloni et al., 2016). As such, the ice sheet's thermal structure at depth could be
estimated by comparing the observed $T_B$ and a simulated $T_B$ derived through microwave emissivity modelling,
including one-dimensional modelling of the ice sheet's temperature profile.  Included in the assumed values for
this modelling are the GHF and the accumulation rate; the sources of greatest uncertainty. This method only
applies in areas of slow flowing ice ($<10$ m yr$^{-1}$), and is optimal in areas of very slow flowing ice ($<5$ m yr$^{-1}$) as

Number: 1     Author: brice     Subject: Inserted Text     Date: 13/04/2020 11:30:53

again if at the PMP, for me it is more a minimum GHF as we don't know whether there is melt or not. If the melt is low, both values are very closed to each other 
[revised manuscript text omitted]

Number: 1     Author: brice     Subject: Inserted Text     Date: 10/04/2020 17:18:13
 You should include the new TCD paper of  Talalay 2020: https://www.the-cryosphere-discuss.net/tc-2020-32/tc-2020-32.pdf

Number: 2     Author: brice     Subject: Underline     Date: 10/04/2020 17:19:47
Dome Fuji is not frozen to the bed, so it is  a minimum GHF

Number: 3     Author: brice     Subject: Inserted Text     Date: 13/04/2020 11:31:34
I strongly suggest to add a few words on machine learning techniques as done in Greenland by Rezvanbehbahani et al. (2017). See before and something like: "Machine learning techniques used to determine the GHF over the Greenland Ice Sheet (Rezvanbehbahani et al., 2017) could be developed for the Antarctic Ice sheet. Up until now, we provide a statistical analysis of basal temperatures and GHF based on the use of different data sets. The use of a Monte Carlo approach, which is based on repeated random sampling to calculate GHF, could bring new perspectives on the data, and in particular on associated uncertainties which would allow us to critically assess our results."

[Figure]

2016). However, (as noted above) local temperature gradients may not be representative of the regional heat flow, as local geology, hydrothermal circulation, and topography can result in localised GHF variability.
[Figure]
 In response, multiple boreholes are required to categorise the regional variation, and topographic effects must be considered and accounted for.

It is also a necessity that thermal modelling of the bedrock temperature profile for the RAID target sites is executed prior to drilling to constrain the penetration depth of low-frequency time variation of temperature. Whilst the

RAID target bedrock borehole depth of ≥25 m is much shallower than the >100 m borehole depth achieved for exposed bedrock (Section 3.1.), the overlying ice sheet insulates the bedrock temperature profile from short duration surface temperature variability (temperature variation penetration depth is dependent on the frequency of the variation and thermal diffusivity of the material; Carslaw and Jaeger, 1959). However, as is considered for

GHF estimates from ice boreholes (Section 3.2.), low-frequency variation in surface temperatures, heat advection, and shear heating will all affect the subglacial temperature profile. Consequently, low-frequency temperature variation must be corrected for, and boreholes are best drilled where the ice is stationary and frozen to the bed (as is applied to ice borehole selection for GHF estimation). By drilling in such sites where glaciological approaches are most effective for GHF estimation, the RAID data will allow validation of GHF estimates for the various englacial temperature methods applied to stationary ice at ice divides (Section 5.). These methods include borehole temperature profiles, subglacial lakes, ice sheet models, and microwave emissivity. It is thus important that the englacial temperature profile is measured in addition to the bedrock temperature gradient.

Beyond bedrock drilling there is lot to be gained from further ice borehole drilling. Firstly, existing data must be evaluated to ensure the methodologies of GHF modelling from borehole temperature profiles are consistent and accurate. This is particularly true for the Dome C borehole, for which the previously published 49.0 mW m$^{-2}$ value (de Mendoza et al., 2016) has been retracted. Future ice boreholes into stationary ice frozen to the bed has the potential to supplement the existing borehole and probe-derived GHF estimates, particularly if the proposed methodology for shallow boreholes can be validated (600 m depth, or the upper 20% of the ice column; Section

3.2.).

**7.2. Geophysical GHF estimates**

[Figure]

Magnetic model
-
seismic model
heat flow
difference,
mWm$^{-2}$

> 30
-10
-20
< -30

Number: 1    Author: brice    Subject: Inserted Text    Date: 10/04/2020 17:33:37
and only where the bed is frozen.

[revised manuscript text omitted]

---

## Referee Comment (RC2) · Anonymous Referee #2 · 20 Apr 2020

Review of Burton-Johnson et al. Geothermal heat flow in Antarctica: current and future directions

Burton-Johnsen et al. review the geological and glaciologic methods for inferring the geothermal flux of Antarctica. This is an important question for a variety of glaciological applications and the understanding of Antarctic geology. The paper in well written and informative. Probably the best description of its utility is that I've already sent it to 3 of my students to help them understand the different methods used to infer geothermal flux.

What I've found most interesting about the paper is the description of the geologic methods. As a glaciologist, these techniques have been difficult to understand and evaluate in the primary papers and I found this paper helpful, but given my specialty,

[Figure]

I am also unable to critically review the geology-based content. It might be worth ensuring that a geologist/geophysicist provides assessment as well.

My only major concern of this manuscript centers on the table of compiled geothermal flux estimates. There has been no significant evaluation of the estimates provided, despite a column entitled "DataQuality". Some sites having multiple different estimates, such as WAIS Divide, with no explanation or evaluation of the difference. To be truly useful, this table needs to be better curated with commentary on why the measurements should or should not be accepted. Because of the huge uncertainties in many (if not all) of the methods, many authors have unwisely justified their own results with erroneous or preliminary interpretations of others. It would be a great service to help remove the confusion about the quality of the different measurements. I realize this is a considerable exercise and understanding the details of each and every measurement is likely beyond what is reasonable for the authors, but even a cursory classification of the confidence in each measurement is hugely helpful. And discrimination/unification of multiple estimates for the same sites seems like a reasonable request.

Minor Comment: The authors are very optimistic about long-wavelength microwave emissivity, which I do not believe is yet warranted. In reviewing Macelloni et al, the authors acknowledge that there was little actual verification, so it is not clear that this technique has added useful information. While a discussion of this technique is useful, it needs a fuller description of the limitations and how difficult they will be to overcome. The sentence on L922-924 reads like a direct funding plea and should be avoided.

Specific comments: L9: I know I'm tilting at windmills here, but "geothermal heat" is redundant. Just "geothermal" is enough.

L13: provide at least one sentence on what you found by reviewing methods and compiling estimates before jumping into future directions

L15: Be specific about how the EAIS is the most sensitive to geothermal flux. The EAIS is not uniformly the most sensitive. For instance, the flux of ice in the Ross Ice

Streams is incredibly sensitive to geothermal flux and basal water which is not true of the vast majority of EAIS.

L16: long-wavelength microwave emissivity has not been sufficiently demonstrated to be useful to warrant a specific bullet point

L25-27: This seems like an overstatement. I would argue the ice-bedrock friction which controls the basal sliding rate is far less constrained and much more important.

L30-33: Provide references

L52: change "lower" to "smaller" that size and position are not confused

L76: I don't think the equation came through in the correction form, or else something else is wrong. There is no integral and lamda is never defined. This whole section is therefore very confusing.85

L85: Geothermal flux is actually not that important to the ice temperature. The accumulation rate and surface temperature are much more important. Once the geothermal flux is sufficient to cause basal melting, the temperature profile is only minorly impacted even for large variations in geotherm flux.

L99: flip -7 and -13 to be consistent with text and ordering

L99: Reword this sentence because an increase from -13 to -7 would not change the basal melt rate since it would still be below freezing. So be more specific about the threshold behavior of variations in geothermal flux.

L106: The PMP effect is incredibly small. The PMP decreases the basal temperature by ∼2C from the thin ice at the coast to the thick ice in the interior wherease the surface temperature decreases by ∼30C. This just isn't a big effect.

L121: change "are" to "can be" because some ice streams, like the Ross ice streams, with very low driving stress don't produce a lot of melt and are appear to be freezing at the bed (Joughin et al., 2004, Melting and freezing beneath the Ross ice streams,

Antarctica).

L122-125: This is a subtle concept, so please articulate what is happening in more detail.

L137-138: Frozen beds are not a prerequisite for deep coring operations. Dome C, Dome Fuji, NGRIP and NEEM are examples of drilling to temperate beds. Potential melting may necessitate clean access or a buffer, but the language should be clear.

L243: Fig 3: This seems like a really strange example to choose. First, there is nothing in the figure which shows what the geothermal flux is. Second, where is the bed? Third, this appears to be a high accumulation rate and shallow ice thickness site, where the temperature profile is dominated by advection, and the confidence in the geothermal flux is low. I'd recommend using either Law Dome (Dahl-Jensen et al., 1999) or Siple Dome (Engelhardt, 2004) as better examples.

Paragraph starting at L247: Maybe I'm just confused, but where is advection in this? This seems appropriate for a material that doesn't flow. If I'm wrong and advection is being considered, please articulate how it fits into the circular frequency.

L267: These three references are all abstracts. I don't think this method will actually work because there is too much memory of past temperature, too much uncertainty in the vertical velocity profile, and too much uncertainty in the thermal conductivity of ice. (also, this will really only provide an lower limit since increasing the geothermal flux will cause basal melting at some point and then the temperature profile because relative insensitive to variations in geothermal flux)

I don't feel qualified to comment on section 4.

L661: I don't like the forward vs. inverse model distinction since many of the "forward" methods are based on inverse methods.

L665: change "ground" to "ice"

L711: Section 5.2. This makes a compelling case for inferring minimum geothermal flux. I'd suggest adding to this section the reverse case: that is, the maximum geothermal flux can be estimated if the ice is known to be frozen to the bed. Raymond Arches are compelling evidence of a frozen bed, which Fudge et al. (2019) used to estimate maximum geothermal flux for two Siple Coast ice rises. Together, the minimum and maximum inferences can be more useful than either alone.

L728: Describe how far from South Pole. There's a lot of discussion of the South Pole lake, which is ~100km away, so the work of Jorden et al. is actually pretty far removed from South Pole.

L736: I think the authors are overly optimistic about this method. In reviewing Macelloni et al., it seems like they were able to infer little besides the ice being warmer with depth. The authors acknowledge that there was little actual verification. While the section should be included, it needs a fuller description of the limitations and how difficult they will be to overcome.

L767: I think you should add a section on using englacial attenuation to determine ice sheet temperature. While published literature has mostly focused on depth-age average values (MacGregor et al., 2015, doi:10.1002/2014JF003418; Schroeder et al., 2016, doi: 10.1017/jog.2016.100), there has been considerable progress with obtaining depth profiles, particularly as multiple radar systems can be cross-compared.

L852: This sentence underestimates what is known about Dome C geothermal flux. The geothermal flux is actually quite well constrained by glaciological modeling (see Parennin et al., 2007). The de Mendoze retracted paper is an incredibly strange reference that is completely outside of the glaciological and ice core communities. I think this is very misleading.

L857: Figure 16: would a log scale make more sense since the 30mW/m2 cutoff seems too small

[Figure]

L861-874: Thank you for this paragraph. It is wonderful to read an insightful critique of the Cure depth technique. I think you are too diplomatic when you write "without being critical of the model itself".

L915-924: I think this section is too optimistic. The last sentence in particular is inappropriate.

L960: "sliding" not "slide"

---

## Author Comment (AC1) · 30 Jul 2020

Dr Alex Burton-Johnson British Antarctic Survey Natural Environment Research Council High Cross, Madingley Road Cambridge CB3 0ET

E-mail: alerto@bas.ac.uk

Dear John Goodge, Thank you for taking the time to provide comments on our manuscript. Whilst these were largely structural, we agree that they will make a large improvement to the coherence of the manuscript in light of its length and the diversity of work included. Our comments are provided below alongside your review. All the best,

Dr Alex Burton-Johnson

[Figure]

J. Goodge jgoodge@d.umn.edu

Nice paper and worthwhile compilation of ideas as well as a look forward. I have a few suggestions, mainly to help improve organization of topics. I wonder about the overall organization of section 4.6, which is about how we make GHF estimates in heterogeneous crust. The opening section 4.6.1 goes into determining heat production from rock samples obtained from exposure, but does not discuss interpretations of GHF offered in these papers. On the other hand, GHF is discussed in sections 4.6.3 and 4.6.4, building on other ways to get at heat production. Seems perhaps better to comment on the implications for GHF from the heat production studies and how this reflects heterogeneities?

- The implications of the studies are now included as a new paragraph at the end of Section 4.6.1.

Further, I understand the distinction between rock outcrop and sampling rocks from moraines, but I wonder if it would make more sense to move up the discussion of glacial moraine materials from the CTM either into section 4.6.1 or perhaps changing that clast section to follow the other as new section 4.6.2? They both relate to determining heat production in rocks.

- Paragraph moved to follow the section on HPE measurement from bedrock.

Also, I suggest changing the title of section 4.6.4 from 'Detrital material' to 'Glacially-derived rock clasts' or something along those lines. For better or worse, 'detrital material' to many people will conjure up detrital minerals from sedimentary deposits or sedimentary rocks, or even sediment itself. In this case, it's an important distinction because we sampled large rock clasts that can be treated analytically just like any rock samples taken from exposure.

- Reworded as suggested.

John Goodge

Please also note the supplement to this comment:
https://tc.copernicus.org/preprints/tc-2020-59/tc-2020-59-AC1-supplement.pdf
* * *

---

## Author Comment (AC2) · 30 Jul 2020

Dr Alex Burton-Johnson British Antarctic Survey Natural Environment Research Council High Cross, Madingley Road Cambridge CB3 0ET

E-mail: alerto@bas.ac.uk

Dear Brice Van Liefferinge, Thank you for taking the time to provide such a helpful and thorough review. You provided many pertinent comments, and their implementation has greatly improved our manuscript. We have addressed all of your points, and list them below alongside your review. All the best,

Dr Alex Burton-Johnson

Brice Van Liefferinge (Referee) bvlieffe@gmail.com

Dear Authors, I enjoyed reading your paper on the comparison of the different GHF estimate methods. This paper provides a great overview of the work done on GHF reconstructions and provides key future directions. All known methods are described and are well supported by explicit examples and references in the manuscript. I really liked section 4 on GHF derived estimates. I think that all the key references are included. I added a few and strongly suggest to add and describe the work of Rezvanbehbahani et al. (2017) on machine learning techniques as done in Greenland. The introduction and conclusion support well the manuscript as do the figures (see specific comments). The language used is appropriate. However, I would suggest the following general changes: a) In the title you use "flow" but I would suggest to use "flux", as well as for the whole manuscript (see specific comments). b) The manuscript is qualitative in a few paragraphs. More quantitative descriptions could be provided such as maximum and minimum GHF of the different data sets, discuss the representativeness of point GHF values, . . . where possible. c) The limitations of the different methods to estimate GHF (ice borehole measurements, model estimates, . . .) are not always discussed. A sentence could be provided for each. E.g. ice borehole measurements provide a minimum GHF value when the base is at the pressure melting point. d) Figure 6 needs to be discussed in more detail in the text. A description of the different data sets used is lacking (see specific comments). e) Section 4.5 is quite long compared to the other subsections, and describes in a lot of details a technique that is not widely in Antarctica because of the lack of measurements. It seems therefore that including specific equations is superfluous, and perhaps the paragraph on its application in Antarctica could then be extended. I attach a detailed review of the paper for the specific line-by-line comments, see attached PDF. All the best, Brice Van Liefferinge

Reply to Main Comments:

a) In the title you use "flow" but I would suggest to use "flux", as well as for the whole manuscript (see specific comments).

- This was a topic of discussion when drafting the SCAR-SERCE White Paper on geothermal heat flow (Burton-Johnson et al., 2020). The community consensus was that "flow" is the correct terminology. "Heat flow" is not limited to the movement of material, but the mechanism of heat transfer (dominantly by conduction when near the Earth's surface). It was highlighted that the two terms are used interchangeably within the scientific literature, but "heat flow" has been established for decades to describe the rate of heat transferred across the surface of Earth per unit area, and is the term used by the International Heat Flow Commission. The most important consideration is to state the units (mw/m2), as we have done. A very important and worthwhile discussion and consideration though, and we thank you for raising it. Hopefully this will bring us closer to a more consistent use of the terms.

b) The manuscript is qualitative in a few paragraphs. More quantitative descriptions could be provided such as maximum and minimum GHF of the different data sets, discuss the representativeness of point GHF values, . . . where possible.

- Maximum and minimum values are not the best way to represent these datasets. Following Burton-Johnson et al. (2017), to be more quantitative we have added probability density plots of the GHF estimates in East and West Antarctica, and added maps of mean and SD for the different geophysically-derived GHF estimates.

c) The limitations of the different methods to estimate GHF (ice borehole measurements, model estimates, . . .) are not always discussed. A sentence could be provided for each. E.g. ice borehole measurements provide a minimum GHF value when the base is at the pressure melting point.

- The advantages and limitations are now summarised in a table.

d) Figure 6 needs to be discussed in more detail in the text. A description of the

different data sets used is lacking (see specific comments).

- The section where each model is discussed has been added to the legend. All of the data coverage shown is discussed in detail in each section.

e) Section 4.5 is quite long compared to the other subsections, and describes in a lot of details a technique that is not widely in Antarctica because of the lack of measurements. It seems therefore that including specific equations is superfluous, and perhaps the paragraph on its application in Antarctica could then be extended.

- Section significantly shortened.

Please also note the supplement to this comment: https://www.the-cryosphere-discuss.net/tc-2020-59/tc-2020-59-RC1-supplement.pdf

Replies to specific comments Supplemental comments, ordered by "Page. Number." E.g. "3.2." is comment two on page three of the supplement.

1.1. This might seem to be a picky comment but it is important to be precise. I would suggest to use "flux" than "flow": heat flow should be reserved for the movement of material while heat flux is a transport of a quantity of energy over time. As in this paper you focus more on the GHF beneath the Ice Sheet (bedrock surface), I would use "flux". Otherwise can you explain the use of "flow"?

- See reply above (Main comments (a) )

1.2. "estimate" (extract sounds like by effort or force)

- Changed.

2.1. present and

- Text added.

2.2. See comment on the Title

- See reply to earlier comment (1.1.)

4.1. I would suggest to add somewhere that ice sheet temperature is also influenced by the ice thickness: as ice acts as an insulator, the greater the ice thickness, the warmer the ice at the base. This is counterbalanced by cold temperature advecting from the surface, itself influenced the accumulation rate.

- Both points now added to this section.

4.2. Whole

- To avoid the confusion with an equal temperature effect through the whole ice sheet thickness, we have decided to exclude this addition.

4.3. Basal

- Added.

4.4. Cross-Out

- Changed

4.5. "increase" ==> from -13°C to 7°C

- Changed.

4.6. and expands the surface area of the bed at the pressure point from 16% to more than 50%.

- Details added.

4.7. Cross-Out

- "Non-uniform" is the correct spelling.

4.8. As you mention surface temperature in this paragraph, I suggest to add a sentence on surface accumulation which can have a strong influence on the basal conditions even in the interior of the ice sheet, and counteract the effect of the GHF (see Fig. 2 Van Liefferinge and Pattyn 2013)

- Accumulation rate now mentioned.

5.1. You should also add the work of Rezvanbehbahani et al. (2017) : Rezvanbehbahani et al. (2017) use for the first time machine learning techniques to derive GHF from relevant geologic features (gravity measurements, magnetic anomaly) and GHF measurements (derived from crustal thickness, rock composition and active thermal feature). S. Rezvanbehbahani, L. A. Stearns, A. Kadivar, J. D. Walker, and C. J. van der Veen. Predicting the geothermal heat flux in Greenland: a machine learning approach. Geophysical Research Letters, 2017. ISSN 1944-8007. doi: 10.1002/2017GL075661.

- Reference added in reply to comment 29.3 (below).

5.2. Cit: Fischer 2013, Climate of the past paper. https://www.clim-past.net/9/2489/2013/

- Citation added.

5.3. Van Liefferinge and not Liefferinge

- Changed

6.1. Curie Point Depth (CPD as in section 4.1 L311)

- Changed

6.2. I suggest to develop in one or two sentences the implications of the Antarctic Ice Sheet like in section 2.3.2

- Apologies, we do not understand this request. This section is discussing how GHF studies inform mantle dynamics research.

6.3. It is a simple suggestion but why not provide a table presenting all the methods used to estimate the GHF with the advantages and disadvantages, to have an overview of all the methods together. This sentence could be extended as well to give an overview of the section's content.

- Summary table of methods added.

7.1. GHF

- Changed

8.1. In the xlsx supplementary material, I guess that in the row "method", borehole means "Boreholes into bedrock" ? If yes can you provide the exhaustive (or estimation) of the number of boreholes into bedrock in Antarctica and cite the SOM.

- Number of data points of each type now added to the "Existing Data" section.

8.2. A key point that is not explained here, is that, when the base of the ice sheet is at the pressure melting point (presence of water), the GHF estimate is a minimum GHF estimate, which means that the GHF can be higher! See also section 5, 5.1

- We agree with the reviewer that if the base of the ice sheet is at the pressure melting point, GHF cannot be estimated from the gradient of temperature alone and only a minimum value of GHF can be estimated. We believe that is clearer to refer in this Section only with studies estimating GHF and we have explicitly stated that one of the requirements is "that the ice sheet has been unequivocally frozen to the bed for long enough that the bedrock and overlying ice sheet have thermally equilibrated"

8.3. add citation

- Citations added

11.1. A: Fox Maule et al., did you use the 2005 version of the data set or the updated one from Purucker ? Based on the figure, you are using the Purucker et al update so please cite: M. Purucker. Geothermal heat flux data set based on low resolution observations collected by the champ satellite between 2000 and 2010, and produced from the mf-6 model following the technique described in fox maule et al.(2005). See http://websrv. cs. umt. edu/isis/index. php, 2013.

- The updated reference is added to the figure and where appropriate in the rest of the

manuscript.

12.1. I would suggest to be very explicit about what each data set is in that figure. e.g. Passalacqua et al., 2017: "radar reflectivity and inverse modelling". Provide a table with the links to the data sets ? (see general comments)

- The section where each model is discussed has been added to the legend.

12.2. derived estimates (to be consistent with the other sub-section title )

- Changed.

13.1. CPD

- Changed.

14.1. . B)

- (B) is used as a reference to the illustration whilst explaining (A) rather than a separate description.

14.2. Cross-Out

- See 14.1

14.3. Highlighted

- See 14.1

15.1. CPD

- Added.

15.2. Cross-Out

- Corrected.

15.3. :

- Corrected.

15.4. bottom of the magnetic source

- Corrected.

15.5. As is, to me, this figure illustrates more the difference between Martos et al. (2017) and the two methods than between seismic and gravity modeling. Why not simply plot the difference between the two methods you describe?

- Whilst the difference between the figures is due to the difference between the two Moho estimates, the figure is to show that there are regions where the Curie depth is deeper than the Moho estimate regardless of the Moho estimation method. Consequently, this is the most appropriate way to present the data.

16.1. derived estimates (see 4.1)

- Reworded.

17.1. Thanks for pointing this difference out !

- Thank you.

17.2. model derived estimates (e.g)

- Reworded.

17.3. as before for the sub-title.

- Reworded.

18.1. Can you provide some numbers on the maximum and minimum GHF ?

- This is dependent on the borehole estimates used.

18.2. Section 4.5 is quite long compared to the other subsections, and describes in a lot of detail a technique that is not widely used in Antarctica because of the lack of measurements. It seems therefore that including specific equations is superfluous, and

perhaps the paragraph on its application in Antarctica could then be extended.

- Section significantly shortened.

21.1. Simplify in one sentence.

- Section re-worded and shortened.

23.1. You use whilst several times, you could remove some of them.

- Alternatives are now used here and elsewhere in the document.

23.2. Underline

- See preceding comment.

23.3. you can also cite: J. W. Goodge, C. M. Fanning, C. M. Fisher, and J. D. Vervoort. Proterozoic crustal evolution of central East Antarctica: Age and isotopic evidence from glacial igneous clasts, and links with australia and laurentia. Precambrian Research, 299:151–176, 2017.

- This paper precedes the Goodge (2018) paper, but discusses what can be determined of the age and composition of the unexposed crust, but does not discuss the heat production implications. To avoid confusion for the reader and to direct them to the relevant paper, we have chosen not to include the earlier paper.

24.1. (Van Liefferinge and Pattyn, 2013).

- Corrected

24.2. As it is difficult to know the melt rate, I would say that this is not the GHF estimate but the minimum value of the GHF estimate or the minimum GHF to reach the pressure melting point.

- We agree with the reviewer that if ice is melting at the base only a minimum value of GHF can be estimated. However, all the methods referred in Sections 5.1 and 5.3 use additional constraints to estimate GHF. In Section 5.2 we explicitly state that only

a minimum value of the GHF can be estimated.

25.1. Van

- Corrected

25.2. Van

- Corrected

25.3. ,

- Corrected

25.4. more than

- Added "at least"

25.5. cite: A. P. Wright and M. J. Siegert. A fourth inventory of Antarctic subglacial lakes. Antarctic Science, 24:659–664, 2012.

- Reference added

26.1. again if at the PMP, for me it is more a minimum GHF as we don't know whether there is melt or not. If the melt is low, both values are very closed to each other

- See response to 24.2

27.1. Van

- Corrected.

29.1. You should include the new TCD paper of Talalay 2020: https://www.the-cryosphere-discuss.net/tc-2020-32/tc-2020-32.pdf

29.2. Dome Fuji is not frozen to the bed, so it is a minimum GHF

- Text added.

29.3. I strongly suggest to add a few words on machine learning techniques as done in Greenland by Rezvanbehbahani et al. (2017). See before and something like: "Machine learning techniques used to determine the GHF over the Greenland Ice Sheet (Rezvanbehbahani et al., 2017) could be developed for the Antarctic Ice sheet. Up until now, we provide a statistical analysis of basal temperatures and GHF based on the use of different data sets. The use of a Monte Carlo approach, which is based on repeated random sampling to calculate GHF, could bring new perspectives on the data, and in particular on associated uncertainties which would allow us to critically assess our results."

- Text added

30.1. and only where the bed is frozen.

- Text added ("where the basal ice is frozen to the bedrock").

33.1. Van

- Corrected.

33.2. I would add that up until now, thermodynamical models are still dependent of GHF estimates (large GHF data sets, borehole temperature measurements, ...) - Text added.

33.3. for local measurements

- Added "for local estimates".

38.1. Please also mention the updated version of this data set by Purucker 2013 which is the available data set now

39.1. Cross-Out

- Corrected.

41.1. Van Liefferinge, B., and Pattyn, F.: ...

- Corrected.

41.2. Van

- Corrected.

Please also note the supplement to this comment:
https://tc.copernicus.org/preprints/tc-2020-59/tc-2020-59-AC2-supplement.pdf

---

## Author Comment (AC3) · 30 Jul 2020

Dr Alex Burton-Johnson British Antarctic Survey Natural Environment Research Council High Cross, Madingley Road Cambridge CB3 0ET

E-mail: alerto@bas.ac.uk

Dear Anonymous Reviewer, Thank you for taking the time to provide such a helpful and thorough review. You provided many pertinent comments, and their implementation has greatly improved our manuscript. We have addressed all of your points, and list them below alongside your review. All the best,

Dr Alex Burton-Johnson

[Figure]

Anonymous Referee #2

Review of Burton-Johnson et al. Geothermal heat flow in Antarctica: current and future directions

Burton-Johnsen et al. review the geological and glaciologic methods for inferring the geothermal flux of Antarctica. This is an important question for a variety of glaciological applications and the understanding of Antarctic geology. The paper in well written and informative. Probably the best description of its utility is that I've already sent it to 3 of my students to help them understand the different methods used to infer geothermal flux.

What I've found most interesting about the paper is the description of the geologic methods. As a glaciologist, these techniques have been difficult to understand and evaluate in the primary papers and I found this paper helpful, but given my specialty, I am also unable to critically review the geology-based content. It might be worth ensuring that a geologist/geophysicist provides assessment as well.

My only major concern of this manuscript centers on the table of compiled geothermal flux estimates. There has been no significant evaluation of the estimates provided, despite a column entitled "DataQuality". Some sites having multiple different estimates, such as WAIS Divide, with no explanation or evaluation of the difference. To be truly useful, this table needs to be better curated with commentary on why the measurements should or should not be accepted. Because of the huge uncertainties in many (if not all) of the methods, many authors have unwisely justified their own results with erroneous or preliminary interpretations of others. It would be a great service to help remove the confusion about the quality of the different measurements. I realize this is a considerable exercise and understanding the details of each and every measurement is likely beyond what is reasonable for the authors, but even a cursory classification of the confidence in each measurement is hugely helpful. And discrimination/unification

of multiple estimates for the same sites seems like a reasonable request.

Minor Comment: The authors are very optimistic about long-wavelength microwave emissivity, which I do not believe is yet warranted. In reviewing Macelloni et al, the authors acknowledge that there was little actual verification, so it is not clear that this technique has added useful information. While a discussion of this technique is useful, it needs a fuller description of the limitations and how difficult they will be to overcome. The sentence on L922-924 reads like a direct funding plea and should be avoided.

Reply to main comments above:

a) "a cursory classification of the confidence in each measurement is hugely helpful. And discrimination/unification of multiple estimates for the same sites seems like a reasonable request."

- A qualitative classification of each estimate in the supplementary table has now been added along with a table of the qualifiers for each classification. These qualifiers are based on literature in the manuscript, so although likely controversial in some instances, are not arbitrarily determined (e.g. 90m as a qualifier for bedrock boreholes is based on the observed depth of surface temperature effects in the DVDP boreholes; Decker, 1974; Decker et al., 1975; Pruss et al., 1974).

b) "The authors are very optimistic about long-wavelength microwave emissivity, which I do not believe is yet warranted."

- Discussion of the caveats of this technique have been expanded.

Replies to specific comments

Specific comments: L9: I know I'm tilting at windmills here, but "geothermal heat" is redundant. Just "geothermal" is enough.

- Whilst "heat" is implicit in "geothermal", "geothermal heat flow" is the full standard terminology used in the literature, and we inherit that convention.

L13: provide at least one sentence on what you found by reviewing methods and compiling estimates before jumping into future directions

- Text added.

L15: Be specific about how the EAIS is the most sensitive to geothermal flux. The EAIS is not uniformly the most sensitive. For instance, the flux of ice in the Ross Ice Streams is incredibly sensitive to geothermal flux and basal water which is not true of the vast majority of EAIS.

- With the removal of the specific comment on microwave emissivity, the EAIS is no longer mentioned.

L16: long-wavelength microwave emissivity has not been sufficiently demonstrated to be useful to warrant a specific bullet point

- Point removed.

L25-27: This seems like an overstatement. I would argue the ice-bedrock friction which controls the basal sliding rate is far less constrained and much more important.

- Emphasis reduced.

L30-33: Provide references

- References added.

L52: change "lower" to "smaller" that size and position are not confused

- Changed.

L76: I don't think the equation came through in the correction form, or else something else is wrong. There is no integral and lamda is never defined. This whole section is therefore very confusing.

- Correct, most of the equation seems to have got lost – Quite embarrassing! This is fixed now.

L85: Geothermal flux is actually not that important to the ice temperature. The accumulation rate and surface temperature are much more important. Once the geothermal flux is sufficient to cause basal melting, the temperature profile is only minorly impacted even for large variations in geotherm flux.

- Sentence reworded. The scale and sensitivity of the ice sheet temperature to GHF variation is discussed in the following paragraphs of this section.

L99: flip -7 and -13 to be consistent with text and ordering

- Corrected.

L99: Reword this sentence because an increase from -13 to -7 would not change the basal melt rate since it would still be below freezing. So be more specific about the threshold behavior of variations in geothermal flux.

- -7°C is the mean temperature (already stated). That the basal melt rate increases is shown by the increase 16% to more than 50% of the basal area exceeding the PMP.

L106: The PMP effect is incredibly small. The PMP decreases the basal temperature by âĹij2C from the thin ice at the coast to the thick ice in the interior wherease the surface temperature decreases by âĹij30C. This just isn't a big effect.

- We agree with the reviewer that the effect of pmp on basal temperature is small. However we are discussing here the sensitivity of basal conditions to GHF. Both references cited show that the sensitivity is larger in the interior of East Antarctica and the reason is that the basal temperature is closer to the PMP. In areas where basal temperature is near PMP, small changes in GHF affects the extension of basal melting and has a stronger effect on the ice sheet. We have rewritten the sentence for clarity.

L121: change "are" to "can be" because some ice streams, like the Ross ice streams, with very low driving stress don't produce a lot of melt and are appear to be freezing at the bed (Joughin et al., 2004, Melting and freezing beneath the Ross ice streams, Antarctica).

- Text changed.

L122-125: This is a subtle concept, so please articulate what is happening in more detail.

- Text added. As with all points in this summary paper, the reader is forwarded to the correct reference.

L137-138: Frozen beds are not a prerequisite for deep coring operations. Dome C, Dome Fuji, NGRIP and NEEM are examples of drilling to temperate beds. Potential melting may necessitate clean access or a buffer, but the language should be clear.

- We agree with the reviewer our comment is focussed on 'Oldest Ice challenge' and not true for every ice core location. We have written the sentence.

L243: Fig 3: This seems like a really strange example to choose. First, there is nothing in the figure which shows what the geothermal flux is. Second, where is the bed? Third, this appears to be a high accumulation rate and shallow ice thickness site, where the temperature profile is dominated by advection, and the confidence in the geothermal flux is low. I'd recommend using either Law Dome (Dahl-Jensen et al., 1999) or Siple Dome (Engelhardt, 2004) as better examples.

- Replaced with the Dahl-Jensen (1999) example from Law Dome.

Paragraph starting at L247: Maybe I'm just confused, but where is advection in this? This seems appropriate for a material that doesn't flow. If I'm wrong and advection is being considered, please articulate how it fits into the circular frequency.

- As stated at the start of this section, the method is applicable where the ice sheet is stationary (frozen to the bed) and thermally equilibrated; hence horizontal advection is minimal. For clarity, this point is now repeated at the start of this section.

L267: These three references are all abstracts. I don't think this method will actually work because there is too much memory of past temperature, too much uncertainty in

the vertical velocity profile, and too much uncertainty in the thermal conductivity of ice. (also, this will really only provide an lower limit since increasing the geothermal flux will cause basal melting at some point and then the temperature profile because relative insensitive to variations in geothermal flux)

- The debate over an estimation of a simplified model 'working' or 'not working' is futile. There is no way around the uncertainties that the reviewer is mentioning and we will only say here that in any study they should be propagated into the estimation uncertainty. In any case, we agree with the reviewer that the estimations rely on a simplified models and this need to be clearer in the paper. We have rewritten the sentence.

I don't feel qualified to comment on section 4.

L661: I don't like the forward vs. inverse model distinction since many of the "forward" methods are based on inverse methods.

- Good point; reworded.

L665: change "ground" to "ice"

- Reworded.

L711: Section 5.2. This makes a compelling case for inferring minimum geothermal flux. I'd suggest adding to this section the reverse case: that is, the maximum geothermal flux can be estimated if the ice is known to be frozen to the bed. Raymond Arches are compelling evidence of a frozen bed, which Fudge et al. (2019) used to estimate maximum geothermal flux for two Siple Coast ice rises. Together, the minimum and maximum inferences can be more useful than either alone.

- Text added.

L728: Describe how far from South Pole. There's a lot of discussion of the South Pole lake, which is áĹij100km away, so the work of Jorden et al. is actually pretty far

removed from South Pole.

- Distance added.

L736: I think the authors are overly optimistic about this method. In reviewing Macelloni et al., it seems like they were able to infer little besides the ice being warmer with depth. The authors acknowledge that there was little actual verification. While the section should be included, it needs a fuller description of the limitations and how difficult they will be to overcome.

- The limitations of the method and the challenges in developing the approach are now expanded in the section on current challenges, Section 7.3 (also responding to comment "L915-924" below).

L767: I think you should add a section on using englacial attenuation to determine ice sheet temperature. While published literature has mostly focused on depth-age average values (MacGregor et al., 2015, doi:10.1002/2014JF003418; Schroeder et al., 2016, doi: 10.1017/jog.2016.100), there has been considerable progress with obtaining depth profiles, particularly as multiple radar systems can be cross-compared.

- The studies on temperature-dependent attenuation by Carter et al (2009) and Schroeder et al (2016) are already discussed in Section 5.1.

L852: This sentence underestimates what is known about Dome C geothermal flux. The geothermal flux is actually quite well constrained by glaciological modeling (see Parennin et al., 2007). The de Mendoze retracted paper is an incredibly strange reference that is completely outside of the glaciological and ice core communities. I think this is very misleading.

- Sentence removed.

L857: Figure 16: would a log scale make more sense since the 30mW/m2 cutoff seems too small - Scale expanded to 60 mW/m2

L861-874: Thank you for this paragraph. It is wonderful to read an insightful critique of the Cure depth technique. I think you are too diplomatic when you write "without being critical of the model itself".

- Thank you.

L915-924: I think this section is too optimistic. The last sentence in particular is inappropriate.

- The last sentence has been removed and the limitations of the method expanded (also addressing point "L736")

L960: "sliding" not "slide"

- Changed.

Please also note the supplement to this comment:
https://tc.copernicus.org/preprints/tc-2020-59/tc-2020-59-AC3-supplement.pdf

---

## Author Response (AR1)

Dr Alex Burton-Johnson
British Antarctic Survey
Natural Environment Research Council
High Cross, Madingley Road
Cambridge
CB3 0ET

E-mail: alerto@bas.ac.uk

Dear Alexander Robinson,

Thank you for reviewing our revisions as Editor. We have addressed your point regarding terminology and include our comments and the manuscript with track changes below.

All the best,

Dr Alex Burton-Johnson

**Editor Decision:** Publish subject to minor revisions (review by editor) (19 Aug 2020) by Alexander Robinson

**Comments to the Author:**

The reviewer comments have been thoroughly addressed, so I believe the manuscript is near-ready for publication. At the moment, I would only suggest the following minor further revision below.

Heat flow versus heat flux (response to Brice Van Liefferinge): It is now clear why you use the term heat flow throughout, which you have justified well. To increase the chances of broad adaption of the 'correct' term, I would suggest adding a footnote, or comment when the term is introduced that explicitly states why you use heat flow. This will hopefully help to encourage others to adapt it in the future.

- A paragraph has been added as a new section (1.2.) discussing the terminology and the justifying the use of "flow" rather than "flux", as in the reply to Reviewer Brice Van Liefferinge.

[revised manuscript text omitted]